# Comparison of induced neurons reveals slower structural and functional maturation in humans than in apes

**Maria Schörnig[1†], Xiangchun Ju[1†], Luise Fast[1], Sebastian Ebert[1], Anne Weigert[1‡], Sabina Kanton[1§], Theresa Schaffer[1], Nael Nadif Kasri[2], Barbara Treutlein[1#], Benjamin Marco Peter[1], Wulf Hevers[1], Elena Taverna[1]\***

[1]Max Planck Institute for Evolutionary Anthropology, Leipzig, Germany; [2]Department of Human Genetics and Department of Cognitive Neuroscience, Donders Institute for Brain, Cognition, and Behavior, Radboudumc, Nijmegen, Netherlands

**\*For correspondence:**
elena_taverna@eva.mpg.de

[†]These authors contributed equally to this work

**Present address:** [‡]University of Leipzig, Medical Department I – Haematology and Cell Therapy, Medical Oncology, Leipzig, Germany; [§]Department of Psychiatry and Behavioral Sciences, Stanford University, Stanford, United States; [#]Biosystems Science and Engineering, ETH Zürich, Basel, Switzerland

**Competing interests:** The authors declare that no competing interests exist.

**Abstract** We generated induced excitatory neurons (iNeurons, iNs) from chimpanzee, bonobo, and human stem cells by expressing the transcription factor neurogenin-2 (NGN2). Single-cell RNA sequencing showed that genes involved in dendrite and synapse development are expressed earlier during iNs maturation in the chimpanzee and bonobo than the human cells. In accordance, during the first 2 weeks of differentiation, chimpanzee and bonobo iNs showed repetitive action potentials and more spontaneous excitatory activity than human iNs, and extended neurites of higher total length. However, the axons of human iNs were slightly longer at 5 weeks of differentiation. The timing of the establishment of neuronal polarity did not differ between the species. Chimpanzee, bonobo, and human neurites eventually reached the same level of structural complexity. Thus, human iNs develop slower than chimpanzee and bonobo iNs, and this difference in timing likely depends on functions downstream of NGN2.

## Introduction

Differences in cognitive abilities between humans and non-human primates are thought to depend on greater numbers of neurons and more complex neural architectures in humans (*Geschwind and Rakic, 2013*; *Herculano-Houzel, 2012*; *Smaers et al., 2017*). Several studies have contributed to the current understanding of the molecular and cellular uniqueness of the primate and, specifically, of the human brain. Compared to other mammals, primate brains contain a greater number of neural stem and progenitor cells that generate neurons (*Florio and Huttner, 2014*; *Borrell and Reillo, 2012*; *Lui et al., 2011*; *Smart et al., 2002*). In addition, some types of neurons differ structurally among primate species (*Elston, 2011*; *DeFelipe, 2011*). For example, human cortical pyramidal neurons have longer and more branched dendrites than chimpanzee pyramidal neurons (*Bianchi et al., 2013a*) and synaptogenesis and development of pyramidal neurons in humans and chimpanzees may be longer than in macaques (*Bianchi et al., 2013b*). Given the role of pyramidal neurons in higher cognitive functions (*Goldman-Rakic, 1999*), a higher degree of connectivity in human neocortex than in other primates could provide a basis for human-specific cognitive abilities (*DeFelipe, 2011*; *Bianchi et al., 2013a*; *Bianchi et al., 2013b*; *Elston et al., 2001*).

Induced pluripotent stem cells (iPSCs) that can be generated from somatic cells have expanded the possibilities to compare neurogenesis among apes (*Otani et al., 2016*; *Mora-Bermúdez et al., 2016*; *Kanton et al., 2019*; *Marchetto et al., 2019*). Recently, human iPSC-derived pyramidal neurons have been found to mature slower than their chimpanzee counterparts both structurally and functionally and to end up having higher dendrite complexity and spine density (*Marchetto et al.,*

*2019*). In addition, upon transplantation into the mouse neocortex human neurons were shown to retain juvenile-like dendritic spines dynamics and to mature both structurally and functionally over a protracted window of time (*Linaro et al., 2019*). These findings are in line with previous studies suggesting that the human brain develops and matures slower than that of closely related primates (*Rakic, 2009*; *Somel et al., 2009*; *Charrier et al., 2012*; *Teffer et al., 2013*) and point to the idea that human neurons and human brain develop with different temporal dynamics. A crucial question remains open: is delayed maturation observed at the tissue/organ level regulated by environmental differences or is a cell-intrinsic property of human neurons? To evaluate possible cell-intrinsic differences in the dynamics of neuronal maturation, we used a direct conversion system in which chimpanzee, bonobo, and human pluripotent stem cells (PSCs) are converted directly into neurons using an NGN2 overexpression system (*Zhang et al., 2013*; *Frega et al., 2017*). This protocol allows to uncouple cell cycle progression of neural progenitors and developmental processes from neuronal maturation. It therefore offers a powerful tool to directly study the intrinsic differences in the dynamics of neuronal maturation.

We followed the temporal differentiation of human and ape induced excitatory neurons (iNeurons, iNs) using single-cell RNA sequencing (scRNAseq), electrophysiology, and morphological analysis of dendritic arborization, and we observed a clear delayed maturation of human iNs compared to chimpanzee at the functional and transcriptional level. The direct conversion paradigm we used allows us to develop further previous findings by finding evidence for a delayed maturation being a cell-intrinsic feature of human neurons. In addition, unlike previous reports describing iNs as a homogenous population of cortical, upper layer neurons, we observed heterogeneous neuronal cell populations with a majority of cortical sensory neurons, further suggesting that the delayed maturation is a general feature of human neurons.

## Results

### Maturation of ape and human induced neurons in vitro

We generated iNeurons from three chimpanzee (SandraA, JoC, ciPS01), one bonobo (BmRNA), and three human (409B2, SC102A, HmRNA) iPSC lines and from one human ESC line (H9) using forced expression of the pan-neurogenic transcription factor neurogenin 2 (NGN 2; *Figure 1A*; *Zhang et al., 2013*; *Frega et al., 2017*). For simplicity, we refer to ESCs and iPSCs as PSCs. We followed their differentiation using molecular, electrophysiological, and morphological approaches for up to 8 weeks, as indicated in *Figure 1A,B*. Single-cell RNA sequencing (scRNAseq) and morphological analyses were performed for two chimpanzee (SandraA and JoC), one bonobo (BmRNA), and three human (409B2, SC102A1, and H9) cell lines. All eight cell lines were used for electrophysiological recordings (note that from now on 'ape' will indicate the combined analysis of chimpanzee and bonobo datasets).

Differentiation was characterized by a downregulation of the stem cell markers *NANOG*, *OCT4*, and *SOX2* in both ape and human cells (*Figure 3—figure supplement 1C*), by a change in cellular morphology and by the extension of neurites (*Figure 1C*). This was followed by expression analyses of genes for synapse organization and axonogenesis (*Figure 3—figure supplement 1D,E*). Chimpanzee, bonobo, and human iNs showed a neuron-like morphology at day 7 (d7) of differentiation and formed a dense network by d14. Neurites were positive for TUJI (beta-III-tubulin, a neuronal marker) starting from d3 of differentiation in apes and humans (*Figure 1—figure supplement 1*).

By the end of the differentiation at d35, both ape and human cells formed networks that were positive for MAP2 (microtubule associated protein-2, marker for mature neurons) and SYN1 (synapsin-1, synaptic vesicle marker; *Figure 1D*). The presence of SYN1-positive puncta suggested that the iNs formed synaptic connections.

We checked for the establishment of axo-dendritic polarity by co-staining for TUJI and neurofilaments, cytoskeletal elements localized in axons (a pan-neurofilament antibody was used, abbreviated as Pan-Neu, see *Supplementary file 2* for details). At d3, TUJI largely colocalizes with neurofilaments, suggesting that the cells were not yet polarized (*Figure 1—figure supplement 2*, high magnification in panels B and C). At d7, the degree of colocalization between TUJI and neurofilament markers decreased, suggesting that the iNs established axo-dendritic polarity (*Figure 1E*, *Figure 1—figure supplement 2*). The pattern of staining of the cytoskeletal components did not

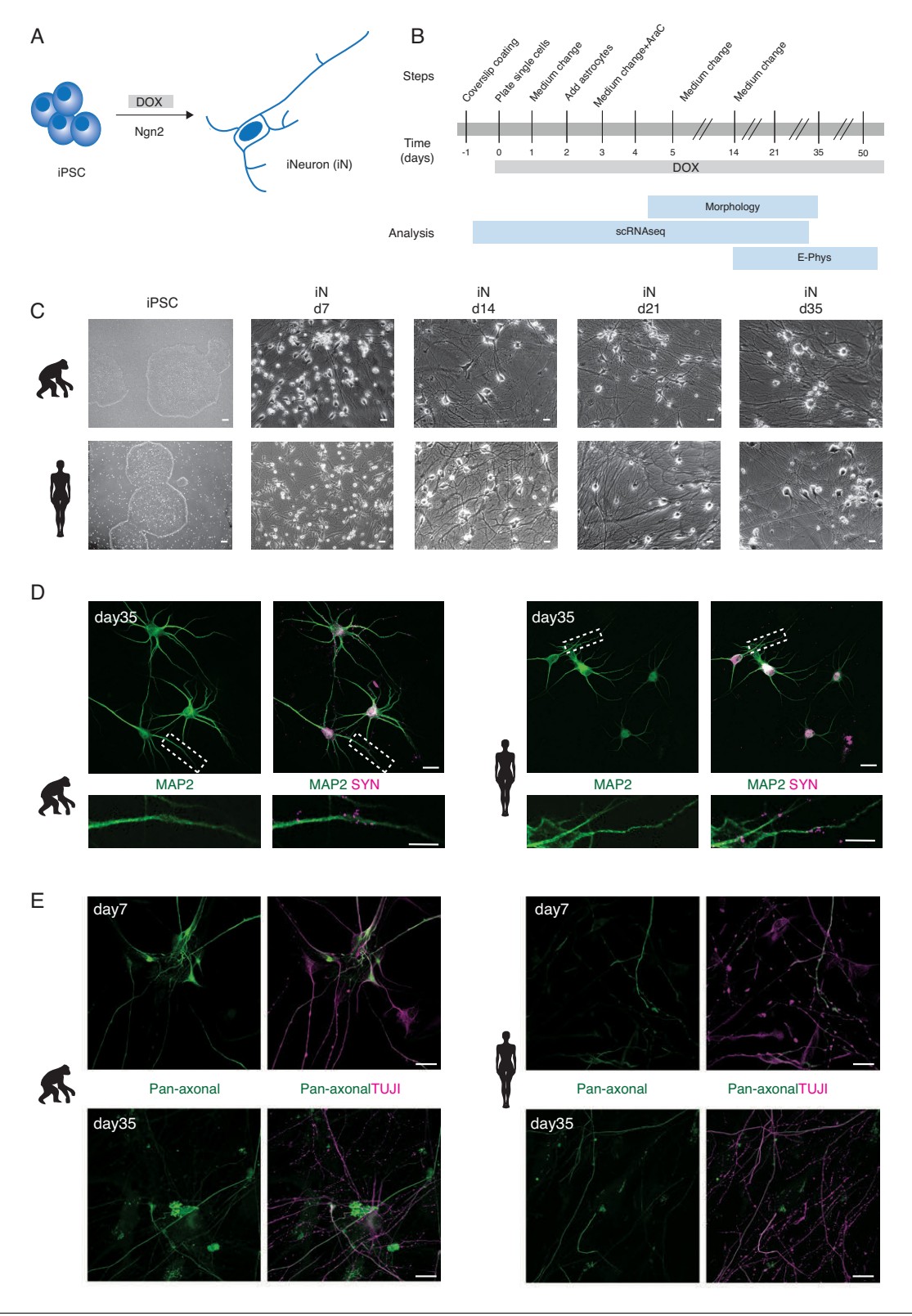

**Figure 1.** Generation of human and ape iNs. NGN2-induced iNs generated from human and ape pluripotent stem cells (A, B) differentiate into mature neurons and express neuronal markers (C, D, E). (A) iNs are generated from iPSCs/ESCs upon DOX-induced overexpression of mouse NGN2. (B) Schematic of the experimental pipeline for iNs structural and functional analysis at different time points. The expression of NGN2 was induced at d0 by adding DOX to the culture medium. iNs were either collected for single-cell RNA sequencing (scRNAseq), used for electrophysiological recordings (E-
*Figure 1 continued on next page*

*Figure 1 continued*

phys), or fixed for morphological analysis (morphology). Single-cell RNA sequencing (scRNAseq) was performed for one chimpanzee (SandraA) and three human (409B2, SC102A1, and H9) cell lines, electrophysiology was performed using all eight cell lines and morphological analyses were done with two chimpanzee (SandraA and JoC), the bonobo (BmRNA), and three human (409B2, SC102A1, and H9) cell lines. (C) Phase-contrast images of iPSCs and iNs maturation for chimpanzee (SandraA) and human (409B2) lines. Scale bars are 10 µm. (D) Mature chimpanzee (SandraA) and human (409B2) iNs express the cytoskeletal marker MAP2 (green) and the pre-synaptic marker SYN1 (magenta). Top: low-magnification view. Scale bars are 20 µm. Bottom: high-magnification view, showing SYN1 puncta in juxtaposition with a MAP2-positive neurite. Scale bars are 10 µm. (E) Distribution of TUJI (TUJI, magenta) and an axonal marker (pan-neurofilament antibody, abbreviated as Pan-Neu, green) in d7 and d35 iNs. Scale bars are 20 µm. We will refer to ape iNs for results where we combined chimpanzee and bonobo iNs for analysis.

The online version of this article includes the following figure supplement(s) for figure 1:

**Figure supplement 1.** iNs express neuronal markers.

**Figure supplement 2.** Polarity establishment in ape and human iNs. Axo-dendritic polarity is established around d7 in chimpanzee (SandraA) and human (409B2) iNs.

differ between apes and humans, suggesting that the timing of axo-dendritic polarity establishment is similar.

We next developed a sparse labeling approach that enables the tracing of individual cells in the dense connected neuronal cultures. This consisted of transfecting iNs with a GFP-encoding plasmid 4 days prior to fixation followed by staining with an axonal marker (Pan-Neu). The majority of iNs (25/26 cells) had a single axon, which is in line with previous findings (*Rhee et al., 2019*). In addition, it was always the longest neurite, which was found to be positive for the axonal marker (n = 26 cells, *Figure 2—figure supplement 1*). Thus, in subsequent analyses, we considered the longest neurite to be the axon.

## Morphological heterogeneity in iN populations

Cells were fixed at d7, 14, 21, and 35 of differentiation and traced using image analysis software (Imaris, see *Supplementary file 1* for details) and quantified using custom software (Materials and methods). For both apes and humans, unipolar, bipolar, and multipolar iNs were observed at all time points analyzed (*Figure 2A–D*). For the apes and humans, 69% and 58% of neurons were multipolar, 26% and 36% bipolar, and 5% and 7% unipolar, respectively (% of all neurons from all time points). We analyzed the bipolar and multipolar iNs separately and disregarded unipolar cells as they may be less mature or less healthy.

## Morphological maturation of ape and human iNs

Both multipolar and bipolar iNs from ape (multipolar iNs: *Figure 2E*; bipolar iNs: *Figure 2—figure supplement 2A*) and human iNs (multipolar iNs: *Figure 2F*; bipolar iNs: *Figure 2—figure supplement 2B*) show an increase in total neurite length over time. At d7 and d14 of differentiation, ape multipolar (*Figure 2G*; $p_{d7}$ = 0.0098; $p_{d14}$ = 0.0215 Mann–Whitney U test) and bipolar (*Figure 2H*; $p_{d7}$ = 0.0278; $p_{d14}$ <0.001 Mann–Whitney U test) iNs have neurite trees of larger total length. At d14, ape cells have more Sholl intersections (intersections of neurites with concentric circles around the cell body) than human cells (*Figure 2K*; $p_{multipolar}$ = 0.0469; *Figure 2L*; $p_{bipolar}$ < 0.001 Mann–Whitney U test). Thus, at early stages of maturation ape, iNs have larger neurite trees than human iNs. At later time points (d21 and d35), total neurite length and number of arborizations did not differ between ape and human iNs.

Human multipolar iNs have slightly longer axons (the longest neurite) at d35 of differentiation compared to ape iNs (*Figure 2I*; $p_{d35}$ = 0.0094 Mann–Whitney U test) (see *Supplementary file 4* for detailed statistics), whereas for other neurites (excluding the longest ones), human neurons have longer total neurite length at d14 of differentiation (*Figure 2M*; $p_{multipolar}$ = 0.0009; *Figure 2N*; $p_{bipolar}$ = 0.447, unpaired T-test), but show no difference at d21 and d35.

## scRNAseq revealed that NGN2 induces cortical and sensory neuron fates

To assess iNs identity and heterogeneity within the culture, chimpanzee (SandraA) and human (409B2) iNs were harvested at d5, 14, 28, and 35 post-induction and used for scRNAseq (10× Genomics). In addition, two human (SC102A1 and H9), one chimpanzee (JoC), and one bonobo

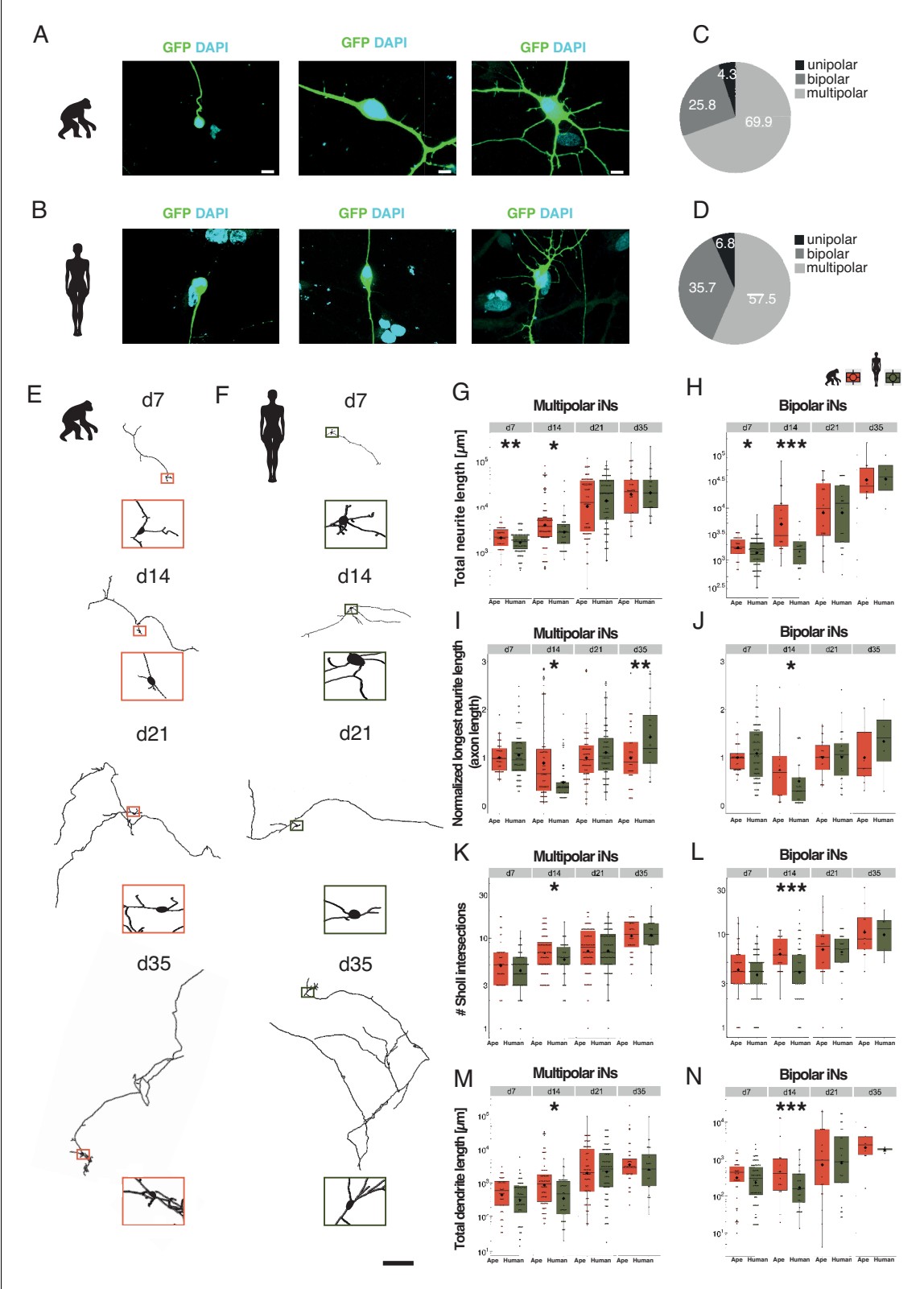

**Figure 2.** Slower morphological maturation of human iNs. iNs were lipofected 4 days prior fixation with a plasmid-expressing cytosolic GFP and fixed at different time points. (A, B) Examples of monopolar (left), bipolar (middle) and multipolar (right) iNs from chimpanzee (SandraA, **A**) and human (409B2, **B**). Cell morphology is highlighted by GFP expression (green); the nuclei are stained with DAPI (cyan). Scale bars are 20 μm. (C, D) Monopolar (black), bipolar (dark gray), and multipolar (gray) iNs in ape (**C**) and human (**D**) expressed as % of total. (E, F) Chimpanzee (**E**, SandraA, orange) and human (**F**,

*Figure 2 continued on next page*

*Figure 2 continued*

409B2, green) multipolar iNs development over time. Zoom-ins show the cell body. Scale bar is 1 mm. (G, H) Total neurite length, expressed in µm for multipolar and bipolar iNs. (G) Apes multipolar iNs show a higher total neurite length at d7 (p=0.0098) and d14 (p=0.0215) compared to human iNs. The scale is logarithmic. (H) Apes bipolar iNs show a higher total neurite length at d7 (p=0.0278) and d14 (p=0.0003). The scale is logarithmic. (I) Relative length of the longest neurite (axon) in apes and human multipolar iNs. Data representation as normalized data against the mean per batch of the ape data in logarithmic scale. Ape iNs show a higher axon length at d14 (p=0.0294) compared to human iNs, according to total neurite length. Human iNs show a higher axon length at d35 (p=0.0094) compared to ape iNs. (J) Relative longest neurite (axon) length in bipolar iNs. Ape iNs show a higher axon length at d14 (p=0.0426). Data representation as normalized data against the mean per batch of the ape data, the scale is logarithmic. (K, L) Total number of Sholl intersections for multipolar (E) and bipolar (F) iNs. Apes iNs show a higher number of Sholl intersections at d14 for both multipolar iNs (p=0.0469) and bipolar (p=0.0008) iNs. The scale is logarithmic. (M, N) Total dendrite length, expressed in microns for multipolar (M) and bipolar (N) iNs. Ape iNs show a higher total dendrite length at d14 for both multipolar (p=0.0009) and bipolar iNs (p=0.04471). The scale is logarithmic. For all graphs, black rhombs represent the mean and black lines the median. Significance score: p<0.05*, p<0.01**, p<0.001***, Mann–Whitney U test. The online version of this article includes the following figure supplement(s) for figure 2:

**Figure supplement 1.** Characterization of axons in iNs.
**Figure supplement 2.** Dynamics of structural maturation in bipolar and multipolar iNs.

(BmRNA) cell lines were harvested at d14 and d35 to assess neuron maturation and cell heterogeneity in further detail. After data processing, we retained 10,111 ape and 25,923 human iNs across the four time points (*Figure 3A–C*). Unsupervised clustering analysis of the combined datasets identified eight clusters of cells based on gene expression patterns (*Figure 3B*). Ape and human cells occurred in all clusters, albeit at different proportions (*Figure 3A*). Progenitor and neuronal cell populations were identified by *SOX2* and *MAP2* expression, respectively. The cortical markers *CUX1* and *BRN2* (*Figure 3C*) were expressed in both cells expressing progenitor markers and neuronal markers. Other cortical markers such as *SATB2*, *TBR1*, or *CTIP2* were detected in a few cells or not detected.

The clusters include three types of neuronal cells: cells expressing progenitor markers (NP: $SOX2^+$, $VIM^+$, $TOP2A^+$, etc., clusters 1 and 2), intermediate cells (expressing both progenitor and neuronal genes, cluster 3), and neurons (*MAP2*, *SYT1*, *GAP43*, etc., clusters 4, 5, 6, and 7). One fibroblast cluster was also identified ($COL1A1^+$ and $TAGLN^+$, etc., cluster 8) (*Figure 3B–C and E*). We could identify three sub-populations of neurons, by differential expression of *TAC1* (cluster 5), *SSTR2* and *GAL* (cluster 6), and *PIEZO* and *GAL* (cluster 7, *Figure 3B*). All three sub-classes were present in similar proportions for all cell lines and species at all time points of iN differentiation (*Figure 3D*). We then compared our data set to the one obtained in a previous study using the single-cell quantitative RT-PCR analysis (Fluidigm) of iNs (*Zhang et al., 2013*). We could identify the expression of almost all markers in all species and cell lines, with the absence of the forebrain marker *FOXG1* and the receptor *GABRB2* (*Figure 3—figure supplement 2*).

The NGN2-induced generation of cells expressing progenitor markers and differentiated neurons is in line with previous findings (*Nehme et al., 2018*) and with what has been reported for ASCL1-induced neuronal cell populations (*Treutlein et al., 2016*).

Based on scRNAseq of the neurons, 54.6% (d5) - 35.4% (d35) ape cells and 80.8% (d5) - 34.3% (d35) human cells were identified as sensory neurons (SNs), when assessed by the expression of the sensory neurons markers *PRPH* (intermediate neurofilament peripherin), neurotrophin receptor tyrosine receptor kinase B (*TRKB*, encoded by *NTRK2*), and homeodomain transcription factors (*POU4F1/BRN3A*) and *PHOX2B* (*Figure 3F*; *D'Autréaux et al., 2011*; *Zou et al., 2012*). Also, a higher correlation of expression levels to sensory neurons (GSE59739) than to other types of neurons (*Figure 3—figure supplement 5C*) including human iPSC-derived motor neurons (GSE133764 and GSE98288), human ES-derived retinal ganglion cells (GSE84639), and hNPC-derived cortical neurons (GSE142670), indicate these cells as sensory neurons that express different combinations of *TAC1* (substance P), galanin (*GAL*), somatostatin receptor 2 (*SSTR2*), and piezo-type mechanosensitive ion channel component 2 (*PIEZO2*) (*Figure 3B,F*), normally found in somatosensory neurons sensitive to mechanical and/or thermal stimuli (*Kerr et al., 2000*; *Li et al., 2016*; *Ranade et al., 2014*). Sodium voltage-gated channel alpha subunit 9 (*SCN9A*) and fibroblast growth factor 13 (*FGF13*), which in combination selectively regulate heat nociception (*Yang et al., 2017*; *Emery et al., 2015*), are expressed in all sensory neuron clusters (*Figure 3F*). Thus, NGN2 can induce the differentiation of PSCs also into nociceptive sensory neurons.

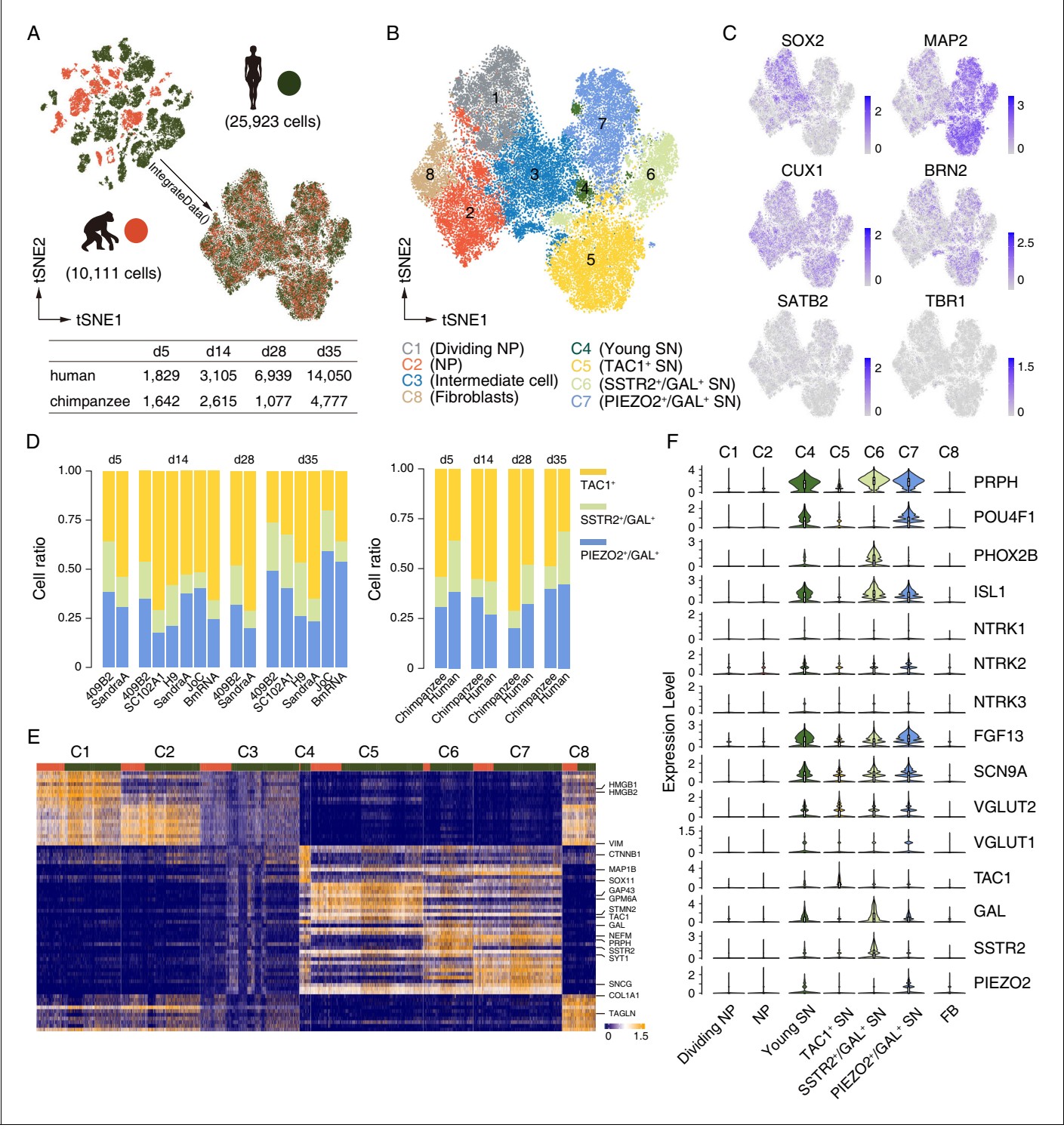

**Figure 3.** scRNAseq analysis reveals that *Nng2*-induced iNs are sensory neurons. Cell identity and heterogeneity were assessed using tSNE combined for all time points for ape (orange) and human (green) cells. (**A**) Filtered cell number for ape and human cells at different time points. (**B**) Identification of 8 different cell clusters by marker gene expression. NP, neural progenitor; SN, sensory neuron. (**C**) Marker gene expression (blue) for progenitor cells (*SOX2*), mature neurons (*MAP2*), and cortical cells (*CUX1, BRN2, SATB2,* and *TRB1*). Scale bars: uncorrected normalized expression. (**D**) Cell ratios of the three neuronal sub-classes of TAC1 (yellow), SSTR2 and GAL (green), and PIEZO and GAL (blue) expressing neurons for all cell lines and species. TAC1+, SSTR2+/GAL+, and PIEZO2+/GAL+ neurons on d5: 54%/36% (chimpanzees/humans), 16%/26%, and 31%/38; TAC1+, SSTR2+/GAL+, and PIEZO2+/GAL+ neurons on d14: 55%/57%, 9%/17%, and 36%/27%; TAC1+, SSTR2+/GAL+, and PIEZO2+/GAL+ neurons on d28: 71%/48%, 9%/20%, and 20%/32%; TAC1+, SSTR2+/GAL+, and PIEZO2+/GAL+ neurons on d35: 49%/31%, 12%/27%, and 40%/42%. (**E**) Heatmap showing scaled expression

*Figure 3 continued on next page*

*Figure 3 continued*

levels of top 10 significantly higher expressed genes in seven cell populations in iNs culture. The main populations we identified are fibroblasts (FB), neural progenitors, and neurons. Top annotation bars: ape (orange) and human (green). (F) The expression levels of sensory neurons markers are shown as a violin plot for neural progenitor's clusters C1 and C2 and for neuron's clusters C4, C5, C6, and C7. Note that the POU homeodomain transcription factors BRN3A (*POU4F1*) and Islet1 (*ISL1*) are highly expressed in a subset of early iNs. All iN clusters show high expression of growth factor FGF13 and voltage-gated sodium channel Nav1.7 (*SCN9A*) that regulate mechano-heat sensation in vivo.

The online version of this article includes the following figure supplement(s) for figure 3:

**Figure supplement 1.** Downregulation of stem cell markers in iNs.
**Figure supplement 2.** Expression of cortical excitatory neuron genes.
**Figure supplement 3.** scRNAseq revealed that NGN2 induces cortical and sensory neuron fates.
**Figure supplement 4.** NGN2 induces cortical sensory neuron fates.
**Figure supplement 5.** Sensory identity of iNs.

From d5 to d35, 47.1% ape and 38.8% human iNs express VGLUT2 (glutamate transporter; *Figure 3F*), while glutamate decarboxylases (GAD1/2, a marker for GABA-ergic cells) is not detected or expressed lowly in either ape or human iNs, suggesting that the iNs are glutamatergic (*Figure 3— figure supplement 2*, see also *Figure 5—figure supplement 1*). Besides expression of sensory markers, we found that 35% (*BRN2*) to 45% (*CUX1*) of ape and 18% (*BRN2*) to 27% (*CUX1*) of human neurons express cortical markers (*Figure 3—figure supplement 3*). Thirteen percentage (*BRN2*) to 21% (*CUX1*) of ape and human neurons express both sensory (*PRHP*) and cortical markers (*BRN2* or *CUX1*). scRNAseq data show that 8% (*BRN2*) to 14% (*CUX1*) of neurons express both sensory (*ISL1*) and cortical markers, suggesting that a subset of neurons might be sensory neurons of the cortex. In spite of the heterogeneity of the cultures (see also *Lin, 2020*), the same cell populations were found in apes and humans, albeit at different proportions, suggesting that NGN2 induces similar neuronal fates in both species.

Our single-cell RNA-Seq data can be explored using our ShinyApp-based browser called iNeuronExplorer (see Materials and methods; *Kanton S, 2021*).

## NGN2 induces cortical sensory neuron fate

The cell identity and the culture heterogeneity of the iNs was further assessed by immunofluoresce for peripheral, cortical, and sensory markers (*Figure 3—figure supplements 4* and *5*). In our hands, only minor proportions (circa 12%) of iNs were found to be positive for Peripherin (PRPH), a marker for peripheral neurons (*Figure 3—figure supplement 4*). We assessed the cortical fate by staining for markers of deep layer neurons (TBR1) and upper layers neurons (CUX1 for layers II–IV and BRN2 for layers II–II) (*Bedogni et al., 2010*; *Oishi et al., 2016*). iNs were negative for TBR1 (*Figure 3—figure supplement 4*). Ninety percentage of the iNs were positive for CUX1, and 50–60% were positive for BRN2 (*Figure 3—figure supplement 4*). Of note, we did not observe any significant preference for a given cell type between the species (*Figure 3—figure supplement 4*). In line with previous reports (*Zhang et al., 2013*; *Frega et al., 2017*), our immunocytochemical data suggest that the vast majority of iNs are upper layers (layers II–IV) cortical neurons in both apes and humans.

The sensory fate was assessed by staining for ISLET1 and NAV1.7, two markers for sensory neurons (*Figure 3—figure supplement 5*). 60% of the iNs were positive for ISLET1 at d21 and d35 (*Figure 3—figure supplement 5A*), and 60–65% were positive for NAV1.7 at d21 (*Figure 3—figure supplement 5B*), with no difference observed between apes and humans. Given that 95% of iNs are positive for CUX1 (*Figure 3—figure supplement 4* and previous paragraph), we conclude that a substantial proportion of iNs are cortical sensory neurons. In line with that, we showed that a proportion of iNs co-express BRN2 and ISLET1 (at the mRNA and protein level, at d21 and d35, see *Figure 3—figure supplement 5A*), further strengthening the idea that a population of iNs are sensory neurons of the cortex (higher-order sensory neurons). The smaller population of sensory iNs not expressing cortical markers is likely to be a population of nociceptive peripheral neurons (*Lin, 2020*).

In summary, based on marker expression at the mRNA and protein level, we conclude that iNs are mainly cortical sensory neurons (for a detailed definition of the criteria we adopted, see Materials

and methods). Furthermore, the same populations are present in similar proportions in apes and humans (*Figure 3—figure supplements 4* and *5*), suggesting that NGN2 induces heterogeneous, yet similar fates in all cell lines and species we analyzed.

## Transcriptional maturation of ape and human iNs

For every time point, we identified genes that are significantly higher expressed in iNs than in the cells expressing progenitor markers (NPs) at the same time point in the same species and iN sub-population and looked for statistical enrichment of these genes among groups of genes assigned to 'Biological Processes' in Gene Ontology (*Bennett and Bushel, 2017*). The human iNs showed similar numbers of higher expressed genes across the different time points. In contrast to the human iNs, the numbers of higher expressed genes observed in ape iNs varied across the time points. Among the iNs sub-clusters of the same species, we observed similar numbers of higher expressed genes at each time point (*Figure 4A*).

To assess the heterogeneity within the iNs sub-clusters, we performed a Pearson's correlation analysis of enriched GO biological processes among iNs sub-clusters for all time points. We found that most iNs sub-clusters of the same species at each time point show high correlation (correlation coefficient > 0.7), indicating a functional similarity among the iN sub-clusters within the same species (*Figure 4B*).

Genes involved in axonogenesis (GO: 0007409) and axon guidance (GO: 0007411) are enriched (p<0.04, binomial test) in both ape and human iNs sub-clusters from d5, although more strongly in the apes (*Figure 4C*). Genes in dendrite development (GO: 0016358) and dendrite morphogenesis (GO: 0048813) are enriched (p<0.05, binomial test) from d5 in the apes, but not in humans (*Figure 4C*). Also, genes involved in synapse organization (GO: 0050808) are enriched from d5 in all ape iNs sub-clusters (p<0.03) and become so from d14 in the human iNs (p<0.02, binomial test) (*Figure 4C*). In addition, genes affecting ion transmembrane transport (GO: 0034220) are enriched from d14 in all ape iNs sub-clusters (p<0.03) but do not become enriched even in d35 human iNs (p>0.33, binomial test). Genes affecting axo−dendritic transport (GO: 0008088) and neurotransmitter secretion (GO: 0007269) are more enriched in ape iNs sub-clusters from d5 and d14, respectively (*Figure 4C*).

Also, we performed an enrichment analysis with the neuron-expressed genes using an expert-curated database of synaptic genes – SynGO (*Koopmans et al., 2019*; *Figure 4D*). In agreement with the results using Gene Ontology, neuron-expressed genes involved in synaptic organization, pre-synaptic process, and trans-synaptic signaling (FDR < 0.05, Fisher exact test) started to be expressed in ape iNs from d5 but only from d14 and d28 in human iNs (*Figure 4E*). Most of the groups of genes involved in post-synaptic process and trans-synaptic signaling were not enriched in human iNs for all time points, while they were enriched in ape iNs from d14 onwards (*Figure 4E* and *Figure 4—figure supplement 2A*). To observe how conserved higher expressed genes among all iNs sub-clusters regulate synaptic functions at each time point, we then focused only on the genes found in all neuronal sub-classes (*Figure 4—figure supplement 1*). In line with the findings in *Figure 4E*, most of the groups of genes involved in post-synaptic process were enriched in ape iNs from d14 onwards and became so in human iNs at d35 (*Figure 4—figure supplement 1*). Of note, the differences between apes and humans are more obvious at early time points. Synaptic gene expression was not enriched in ape or human cells expressing progenitor markers when compared to undifferentiated iPSCs (*Figure 4—figure supplement 2B*), suggesting that these changes are not caused by NGN2 expression per se but as part of the induced neuronal maturation.

Thus, after NGN2 induction, genes associated with dendrites and synapses become prominently expressed later in human iNs than in ape iNs (*Figure 4*).

## Intrinsic passive electrophysiological properties of ape and human iNs

We determined resting-membrane potential (Vrmp), cell-input resistances (Rcell), cell capacitance (Ccell), and the resulting time constant (tau = R*C; see Materials and methods for details) in 144 ape and 155 human iNs at 0–8 weeks after induction of NGN2 expression.

Vrmp was stable over time and similar between species (*Figure 5—figure supplement 1A*; d2 - d > 49, two-way Anova (2WA) $p_{day}$=0.44; $p_{spec}$=0.78; $p_{day*spec}$=0.051). Rcell only increased over the very first days of differentiation and then decreased over time as expected from maturing neuronal

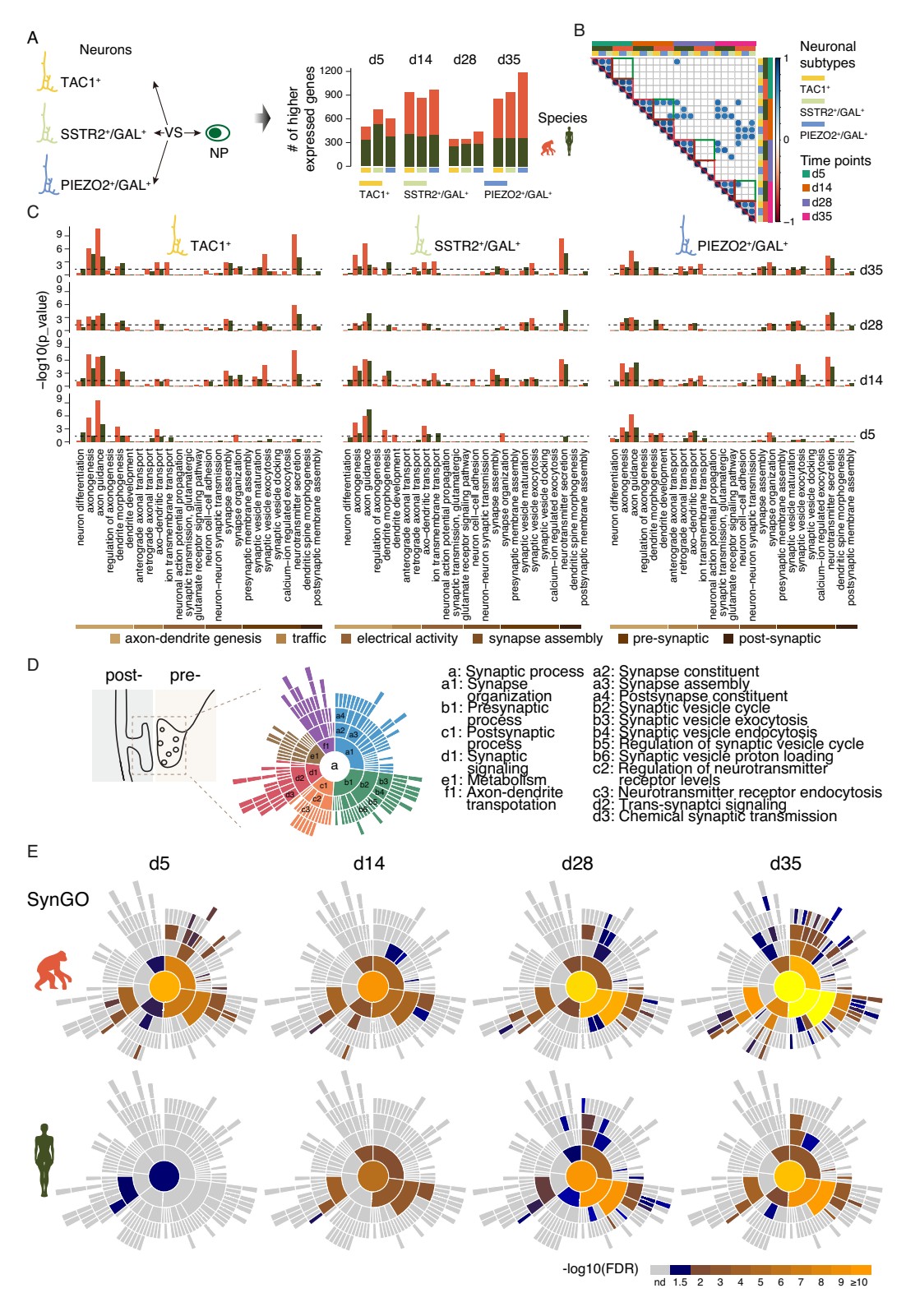

**Figure 4.** scRNAseq analysis of synaptic maturation in ape and human iNs. (**A**) Higher expressed genes in ape and human iN sub-classes compared with corresponding NP of the same species at multiple time points. Left panel, schematics of cell populations for the identification of differentially expressed genes; right panel, bar plot of numbers of higher expressed genes in ape and human iNs. (**B**) Pearson correlation of enriched GO terms in iNs sub types. Circles represent correlation coefficient larger than 0.7. Red triangles indicate correlations between sub types of same species. Green

*Figure 4 continued on next page*

*Figure 4 continued*

boxes indicate correlations within the same sub types but between different species. (**C**) Categories of GO terms on biological processes (BP) enriched with higher expressed genes in ape (orange) and human (green) iN sub-populations at each time point. Reference dashed line (blue) represents p-value of 0.05 (binomial test). (**D**) Left, schematic representation of synaptic parts depicted by SynGO ontology analysis. Right, six top levels (a1–f1) and selected sub-classifiers of SynGO BP terms. (a, synaptic process). (**E**) Sunburst plots of enriched synaptic GO terms in developing ape (top) and human (bottom) iNs. Bottom, the log10-transformed FDR-corrected p value per ontology term is visualized for BP. Note that the bottom BP sunburst plots are aligned with the top-right one.

The online version of this article includes the following figure supplement(s) for figure 4:

**Figure supplement 1.** scRNAseq analysis of synaptic maturation in ape and human iNs.

**Figure supplement 2.** Transcriptional neoteny of synaptic genes in human iNs.

cells (*Figure 5—figure supplement 1B*) with little differences between ape and human (2WA: $p_{day}$=1.8E-13; $p_{spec}$=0.18; $p_{day*spec}$=0.23). As expected, Ccell increased over time (*Figure 5—figure supplement 1A*). While the increase occurred to a similar extent in both groups from about 20 pF– 30 pF and above, the human cells were always slightly but significantly lower than the apes. (*Figure 5—figure supplement 1A*; 2WA: $p_{day}$=7.45E-21; $p_{spec}$=0.02; $p_{day*spec}$=0.12). The resulting time constant R*C of the cells were comparatively stable at about 16 ms (ape: 17.4 ms, human: 15.3 ms) and similar to values previously reported for human (or other) neuronal cells (*Eyal et al., 2016*; *Figure 5—figure supplement 1A*). However, mainly at d14–16, the ape cells had higher values than the human cells mainly due to their high Rcell and low Ccell, respectively (2WA: $p_{day}$=0,17; $p_{spec}$=0.08; $p_{day*spec}$=0.023) (see *Supplementary file 3* for details).

Thus, the differentiation of ape and human iNs is accompanied by maturation of their basic electrical properties in a way consistent with what has been observed for other in vitro and in vivo neuronal systems (*Moody and Bosma, 2005*).

## Active electrophysiological properties of ape and human iNs

Using the whole-cell current-clamp technique, we found that upon induced depolarization, iNs showed single action potentials (APs) as early as day 2 (d2) of differentiation albeit with peak voltages hardly crossing 0 mV (*Figure 5*). Ape iNs showed repetitive action potentials robustly already at d14–16, with its maximal medial frequency increasing slightly over differentiation time (*Figure 5A, B*). In contrast, upon induced depolarization, human iNs fired mainly single APs until d21, after which the ability to maintain repetitive firing became established also in the human lines (*Figure 5A,B*; 2WA: $p_{day}$=1.98E-24; $p_{spec}$=2.96E-6; $p_{day*spec}$=0.15; n=144 ape/156 human iNs). scRNAseq analysis revealed that mRNAs for sodium, potassium, and calcium channels are expressed in both ape and human iNs (*Figure 5—figure supplement 1B*), with no species-specific differences.

We next measured spontaneous excitatory post-synaptic currents (sEPSCs; excitatory nature verified by using kynurenic acid during recordings [*Figure 5—figure supplement 1C*; see also scRNA-seq *Figure 4—figure supplement 2*]) as a proxy for the establishment of functional synapses and cell–cell communication. The ape iNs developed spontaneous activity after 2–3 weeks of differentiation. In contrast, the majority of human iNs were silent at these times (*Figure 5C,D*; 2WA: $p_{day}$=0.053; $p_{spec}$=0.0035; $p_{day*spec}$=0.071; n = 132 ape/135 human iNs; see *Supplementary file 3* for details). sEPSCs became frequent in human iNs only after 3–4 weeks. Before d14, we did not observe any spontaneous synaptic activity.

## Discussion

In this work, we study the differences in neuronal maturation between apes and humans by using a direct conversion protocol, in which iNs are generated by forced expression of NGN2 in ape and human PSCs. We confirm and extend previous findings by showing here that NGN2 expression induces not only cortical fate (*Zhang et al., 2013*; *Frega et al., 2017*), but also sensory fate in both apes and humans. Furthermore, we find that human iNs mature slower than ape iNs from a structural, transcriptional, and functional point of view. To make our data accessible to the neuroscience community, we provide a ShinyApp-based web browser for data exploration (*Kanton S, 2021*).

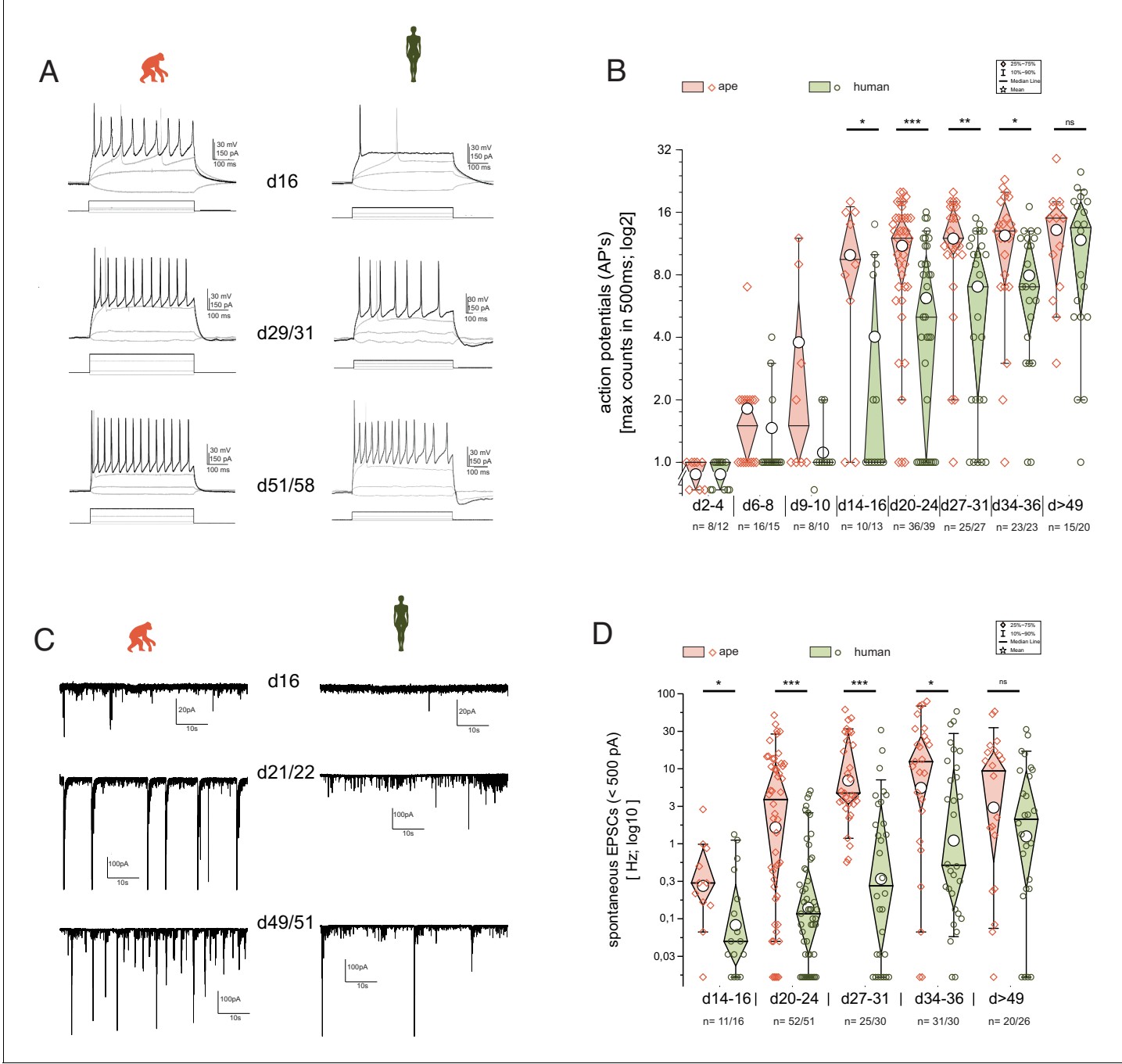

**Figure 5.** Functional development is delayed in human iNs. The development of repetitive action potentials (APs) and spontaneous excitatory post-synaptic potentials (sEPSCs) during iNs differentiation was detected using patch-clamp recordings. (A) Voltage responses of individual cells of ape and human to depolarization by current injection at d16, d29–31, and d51–58. Given are the traces with the maximal inducible numbers of APs (black; in gray the responses to −10 to +10 pA of current injection and the trace with the first AP occurring are shown as comparisons). (B) Boxplots of the maximal AP counts (log₂ scale). The maximal inducible number of APs in each cell was determined and grouped according to their developmental stage representing app. <1, >1, 2, 3, 4, 5, and more than 7 weeks after induction of NGN2. iNs at d2–4 hardly showed action potentials (APs) and if at all only single APs with reduced amplitude. At d6–d10, iNs showed single APs, with some ape iNs showing double or the very first repetitive APs. After 2 weeks of differentiation Ape cells can maintain repetitive APs with a further increase during the following weeks. Human cells after 2 weeks of differentiation fire mainly single APs. Their ability to maintain repetitive firing developed only in the following weeks. (C) Current responses at −80 mV over a time period of 60 s reveal spontaneous synaptic activity. Spontaneous synaptic activity is observed in ape cell lines regularly after 3 weeks of differentiation and cells without are rarely observed at later time points. In humans however, we observed at early differentiation only rarely spontaneous activity which came up mainly following 4 weeks of differentiation. (D) Boxplots of the observed frequencies of sEPSCs in ape and human.

*Figure 5 continued on next page*

*Figure 5 continued*

The number of spontaneous events were counted over 60 s and used for further analysis after $\log_{10}$ transformation to account for their distribution.
Significance score: p<0.05*, p<0.01**, p<0.001***, unpaired T-test with Welch correction.
The online version of this article includes the following figure supplement(s) for figure 5:

**Figure supplement 1.** Basic electrical properties of apes and human iNs.

## NGN2 induces heterogeneous neuronal fates

A surprising finding in our study was that NGN2 induces heterogeneous, yet similar fates in ape and human iNs. The heterogeneity of iNs cultures in human has been so far under-estimated and was recently revealed using the power and resolution of scRNAseq (*Lin, 2020*). Cell heterogeneity poses challenges when using human iPSC cells: for therapeutical applications, there is an obvious need for perfectly controllable and defined cell populations (*Lin, 2020*). However, the heterogeneity can be beneficial for basic research in evolutionary neurobiology, as it allows modeling in a dish several neuronal fates and features, of which only a few are expected to show an evolutionary-relevant difference. In addition, the comparison of different neuronal fates maturing in the same dish offers a valuable internal control when looking for features shared by different cell types that can in turn be regarded as core components/biological processes showing relevant changes between species.

## Evolutionary aspects of neuronal maturation

Previous studies showed neoteny of human neurons compared to apes. Whether the neoteny is regulated by cell-extrinsic or neuron-intrinsic mechanisms is currently unknown. This question cannot be easily addressed using organoids, cell transplantation, or heterochronic cultures, as these are complex systems, where multiple events take place at the time: cycling stem cells give to neurons in an heterochronous way, newborn neurons migrate and mature, while showing robust cell–cell interactions that might regulate and influence their behavior. Despite its possible limitations, the use of iNs allows to uncouple early developmental events such as cell cycle progression, proliferation, and differentiation from the intrinsic neuronal maturation properties, thus offering the unique opportunity to directly study the cell-intrinsic and evolutionary aspects of neuronal maturation.

Our data show that several fundamental properties of neurons, for example the establishment of axo-dendritic polarity, onset of expression of voltage-gated channels, and basic electrical properties, show little or no differences between apes and humans. However, synapse-associated genes become expressed later in human neurons and electrical activity within cells (APs) and between cells (sEPSCs) appear later in the human neurons. In line with that, comparisons of human, chimpanzee, and macaque post-mortem brains have shown that expression changes of genes associated with synaptic functions and spine formation occur later in humans than in the other primates (*Lui et al., 2011*; *Somel et al., 2009*; *Charrier et al., 2012*; *Petanjek et al., 2011*). iNs morphological maturation follows a similar pattern where until d14 post-induction ape neurons have more complex and longer neurites, a difference that disappears by d21.

A slower neuronal development of neuronal gene expression in humans than in apes has been previously described in organoids and at the organ level (*Kanton et al., 2019*; *Somel et al., 2009*). Previous studies in cortical neurons made use of a developmental systems, in which pyramidal neurons derived from iPSC via neural progenitors (NPs) without the forced expression of NGN2 (*Otani et al., 2016*; *Marchetto et al., 2019*). Our data using the iN direct conversion protocol show that the later onset of neuron differentiation and maturation does not only occur in whole organs or complex differentiation systems but happens also in cortical sensory neurons as a downstream effect of NGN2-induced differentiation, strongly suggesting that the delayed maturation is a cell-intrinsic property of human neurons. Of note, sensory neurons are interesting from an evolutionary point of view, as the development and evolution of working memory in human is linked to a higher integration of sensory functions in the human prefrontal cortex (*Smaers et al., 2017*; *Lara and Wallis, 2015*; *Semendeferi et al., 2001*; *Carlén, 2017*).

Which are the reason(s) underlying the transcriptional and functional delay we observed? The synaptic genes expressed by all the neuron sub-classes (core-synaptic genes) show a clear transcriptional delay at early time points (d5). At later time points (d14–21), the transcriptional differences in core-synaptic genes are milder, although the functional differences are still dramatic (see EPSC

frequency). This suggests to us that the maturation of synaptic contacts and communication might be driven by two different mechanisms: a transcriptional driver at early time points, setting iNs apart transcriptionally and morphologically, and a post-transcriptional driver at later time points, setting iNs apart functionally. Although this hypothesis has not been tested experimentally in our study and awaits validation, the idea as such is compatible with several mechanisms governing synapse function being post-transcriptional and post-translational (*Klann and Dever, 2004*; *Sossin and Lacaille, 2010*; *Kelleher et al., 2004*). In addition, an activity-dependent transcriptional and translational regulation (*Ghiani et al., 2007*; *Buffington et al., 2014*) might contribute and possibly amplify the differences between ape and human neuronal maturation at the functional level. At the molecular level, a possible driver for such differences is Slit-Robo Rho GTPase activating protein 2 (SRGAP2) (*Charrier et al., 2012*), wherein three duplication events have occurred on the modern human lineage (*Dennis et al., 2012*). Expression of the human-specific form SRGAP2C variant leads to neoteny in spine maturation and might contribute to human brain evolution (*Charrier et al., 2012*).

The delayed development and maturation of human neurons may have several possible consequences for the organization and function in the adult brain. The later onset of neurite outgrowth shown here and by others (*Otani et al., 2016*; *Marchetto et al., 2019*; *Petanjek et al., 2011*; *Linaro et al., 2019*) has been speculated to result in more complex dendritic trees (*Bianchi et al., 2013a*; *Elston et al., 2001*; *Marchetto et al., 2019*). Interestingly, at d35, we find that the axons of the human multipolar neurons are slightly longer than in the ape neurons. Further work is required to understand whether longer axons allow neurons to develop further and establish more contacts with other neurons. Of note, longer axons would obviously make contacts over greater distances in the brain possible. Since the human brain is three times bigger than the ape brains, in the future it will be interesting to investigate whether axons may be longer in human brains than in ape brains and whether that may affect the extent of neuronal projections in human brains.

# Materials and methods

### Key resources table

| Reagent type (species) or resource | Designation | Source or reference | Identifiers | Additional information |
|---|---|---|---|---|
| Cell line (*Homo sapiens*) | Human iPS cell line | Riken BRC Cellbank | hiPS-409-B2 | Female |
| Cell line (*Homo sapiens*) | Human iPS cell line stable transfected with Ngn2 and rttA | This paper | hiPS-409-B2_Ngn2 | Generated for this study. See Materials and methods. |
| Cell line (*Homo sapiens*) | Human iPS cell line | Systems Biosciences | SC102A-1 | Male |
| Cell line (*Homo sapiens*) | Human iPS cell line stable transfected with Ngn2 and rttA | This paper | SC102A-1_Ngn2 | Generated for this study. See Materials and methods. |
| Cell line (*Homo sapiens*) | Human iPS cell line | This paper | HmRNA | Generated for this study. See Materials and methods. Female Reprogramming by mRNA |
| Cell line (*Homo sapiens*) | Human iPS cell line stable transfected with Ngn2 and rttA | This paper | HmRNA_Ngn2 | Generated for this study. See Materials and methods. |
| Cell line (*Pan troglodytes*) | Chimpanzee iPS cell line | Generated in a previous study (*Mora-Bermúdez et al., 2016*) | Sandra_A | Female |

*Continued on next page*

*Continued*

| Reagent type (species) or resource | Designation | Source or reference | Identifiers | Additional information |
|---|---|---|---|---|
| Cell line (*Pan troglodytes*) | Chimpanzee iPS cell line stable transfected with Ngn2 and rttA | This paper | Sandra_A_Ngn2 | Generated for this study. See Materials and methods. |
| Cell line (*Pan troglodytes*) | Chimpanzee iPS cell line | Generated in a previous study (*Mora-Bermúdez et al., 2016*) | Jo_C | Male |
| Cell line (*Pan troglodytes*) | Chimpanzee iPS cell line stable transfected with Ngn2 and rttA | This paper | Jo_C_Ngn2 | Generated for this study. See Materials and methods. |
| Cell line (*Pan troglodytes*) | Chimpanzee iPS cell line | Provided by the Max-Delbrück-Centrum für Molekulare Medizin | ciPS01 Chimp male iPSC Sendai CL5 | Male |
| Cell line (*Pan troglodytes*) | Chimpanzee iPS cell line stable transfected with Ngn2 and rttA | This paper | ciPS01_Ngn2 | Generated for this study. See Materials and methods. |
| Cell line (*Pan paniscus*) | Bonobo iPS cell line | Generated for a previous study (*Kanton et al., 2019*) | BmRNA | Female Reprogramming by mRNA |
| Cell line (*Pan paniscus*) | Bonobo iPS cell line stable transfected with Ngn2 and rttA | This paper | BmRNA_Ngn2 | Generated for this study. See Materials and methods. |
| Antibody | MAP2 chicken polyclonal antibody | Invitrogen | PA1-16751 | Dilution (1:1000) |
| Antibody | SYN1/2 guinea pig polyclonal antibody | Synaptic Systems | 106004 | Dilution (1:1000) |
| Antibody | TUBB3 mouse monoclonal antibody | BioLegend | 801202 | Dilution (1:1000) Coupled to AlexaFlour 488 or Alexa Flour 555 |
| Antibody | Purified anit-Neurofilament Marker (pan-axonal, cocktail) mouse monoclonal antibody | BioLegend | 8379074 | Dilution (1:400) Clone: SMI312 |
| Antibody | ISL1 mouse monoclonal antibody | Invitrogen | MA5-15515 | Dilution (1:1000) |
| Antibody | PRPH chicken polyclonal antibody | Abcam | ab39374 | Dilution (1:1000) |
| Antibody | BRN2 goat polyclonal antibody | Santa Cruz | sc-6029 | Dilution (1:100) |
| Antibody | TBR1 rabbit polyclonal antibody | Abcam | Ab31940 | Dilution (1:200) |
| Antibody | CUX1 mouse monoclonal antibody | Santa Cruz | sc-514008 | Dilution (1:500) |
| Recombinant DNA reagent | pmax GFP (plasmid) | Lonza | D-00072 | |

*Continued on next page*

*Continued*

| Reagent type (species) or resource | Designation | Source or reference | Identifiers | Additional information |
|---|---|---|---|---|
| Recombinant DNA reagent | pLVX-EF1α-(Tet-On-Advanced)-IRES-G418(R) (plasmid) | Provided by Nael Nadif Kasri (*Frega et al., 2017*) | | This vector encodes a Tet-On Advanced transactivator under control of a constitutive EF1α promoter and confers resistance to the antibiotic G418. |
| Recombinant DNA reagent | pLVX-(TRE-thight)-(MOUSE) Ngn2-PGK-Puromycin(R) (plasmid) | Provided by Nael Nadif Kasri (*Frega et al., 2017*) | | This vector encodes the gene for murine neurogenin-2 under control of a Tet-controlled promoter and the puromycin resistance gene under control of a constitutive PGK promoter. |
| Commercial assay or kit | Stem MACS mRNA transfection Kit | Miltenyi Biotec | 130-104-463 | |
| Commercial assay or kit | Human Pluripotent Stem Cell 3 Colour | Immunohistochemistry Kit | R and D Systems1 | SC021 |
| Commercial assay or kit | Human Pluripotent Stem Cell Functional Identification Kit | R and D Systems | SC027B | |
| Commercial assay or kit | StemMACS Trilineage Differentiation Kit | Miltenyi Biotec | 130-115-660 | |
| Software, algorithm | Imaris 9.5 | Bitplane | | |
| Software, algorithm | Origin | OriginLab version 2018-2019b | | |
| Software, algorithm | Patch and Fitmaster software | HEKA version 2.9x | | |

*Continued on next page*

*Continued*

| Reagent type (species) or resource | Designation | Source or reference | Identifiers | Additional information |
|---|---|---|---|---|
| Software, algorithm | Custom script to analyse neuron morphology based on Imaris raw files (.hoc) | This paper (*Peter and Schörnig, 2020*) | | For quantification, we exported Sholl intersections and .hoc files, as they contain information about both the neurite length and the branching pattern. HOC files were analyzed using a custom script. See Materials and methods. (*Peter and Schörnig, 2020*) |
| Software, algorithm | R 3.5.1 | R | | |

## Generation of rtTA/*Ngn2*-positive PSC lines

For this study, we used three human (hiPS409-B2, SC102A1, HmRNA), three chimpanzee (SandraA, JoC, ciPS01), and one bonobo (BmRNA) iPS cell lines and one additional human ES cell line (H9). The human iPSC lines hiPS409-B2 and SC102A1 were purchased from the Riken BRC Cellbank and System Biosciences, respectively. The human iPSC line HmRNA (generated in this study) was reprogrammed from human dermal fibroblasts using the StemMACS mRNA transfection kit. The cell line was validated for pluripotency markers by immunohistochemical staining using the Human Pluripotent Stem Cell 3-Colour Immunohistochemistry Kit and were differentiated into the three different germ layers using the Human Pluripotent Stem Cell Functional Identification Kit and StemMACS Trilineage Differentiation Kit. Karyotyping was carried out using Giemsa banding at the Stem Cell Engineering facility, a core facility of CMCB at Technische Universität Dresden. Karyotypes were found to be normal. The human ES cell line H9 was purchased from WiCell. The chimpanzee iPSC lines SandraA and JoC as well as the bonobo iPSCs line BmRNA were generated in a previous study (*Kanton et al., 2019*). The chimpanzee iPSCs ciPS01 line was provided by the Max-Delbrück-Centrum für Molekulare Medizin, Berlin.

The rtTA/*Ngn2*-positive iPSCs/ESCs hiPS409-B2_*Ngn2*, SandraA_*Ngn2*, BmRNA_*Ngn2*, H9_*Ngn2*, SC102A1_*Ngn2*, HmRNA_*Ngn2*, ciPS01_*Ngn2*, and JoC_*Ngn2* were generated using lentiviral vectors to stably integrate the transgenes into the genome of the PSCs as previously described by *Frega et al., 2017*.

We used the human lines hiPS409-B2_*Ngn2*, H9_*Ngn2*, and SC102A1_*Ngn2*, the chimpanzee lines JoC_*Ngn2* and SandraA_*Ngn2*, and the bonobo line BmRNA_*Ngn2* for the scRNAseq and morphological experiments. For the electrophysiological experiments, we used additionally the human iPS cell line HmRNA_*Ngn2* and the chimpanzee cell line ciPS01_*Ngn2*.

## Culturing of PSC lines

Human, chimpanzee, and bonobo rtTA/*Ngn2*-positive PSCs were cultured in six-well cell culture plates coated with Matrigel (1:100 dilution in knockout DMEM/F12) and maintained in mTeSRone with supplement and antibiotics (dilution Pen/Strep 1:200, G418 1:1000, and Puromycin 1:2000) at 37 °C and 5% $CO_2$. The medium was refreshed daily.

To maintain the culture, PSC colonies with a round and densely packed morphology were either picked and seeded into wells containing fresh medium or passaged once they reached a confluency of approximately 80%. For passaging, the cells were washed with DPBS before incubation with 1 ml

EDTA (1:1000 dilution in PBS) for 5 min at 37°C to detach the cells from the culture dish. EDTA was aspirated and cells were detached with 1 ml medium. After splitting, the cell suspension was transferred into new wells containing fresh mTeSRone with supplement. Rock Inhibitor (1:1000 dilution) was added for the first 24 hr of the culture. Our cultures were regularly controlled for mycoplasma.

## Cryopreservation of PSCs

Cells were collected and prepared for freezing by washing them with DPBS twice before incubation with 1 ml TrypLE Express for 5 min at 37°C. Cells were washed from the culture dish with 3 ml DMEM/F12 and centrifuged at 200 × g for 5 min. The pellet was resuspended in 500 µl mFreSR, transferred into a cryogenic vial, and gradually frozen to –80°C in an isopropanol-filled freezing box.

To recover the cells, the vial was quickly thawed in a water bath at 37°C. The cell suspension was transferred into 5 ml DMEM/F12 and centrifuged at 200 × g for 5 min. The cell pellet was resuspended in 2 ml medium and transferred into fresh wells containing mTeSRone with supplement and Rock Inhibitor (1:1000 dilution) was added for the first 24 hr of culture.

## Differentiation of rTTA/*Ngn*2-positive PSCs to iNeurons

Human, chimpanzee, and bonobo PSCs were differentiated into induced Neurons (iNs) according to *Frega et al., 2017*. Briefly, we plated iPSCs or ESCs as single cells and initiated the neuronal differentiation by doxycycline induction of NGN2. We pre-coated acid-treated coverslips with 50 µg/ml PLO in borate buffer for 3 hr at 37°C. The coverslips were washed three times with sterile MQ $H_2O$ and coated with 20 µg/ml mouse laminin in DMEM/F12 over night at 4°C. Cells were plated as single cells in mTeSR1 medium supplemented with 1% P/S, 4 µg/ml doxycycline, and ROCK inhibitor (1:1000 dilution). The culture medium was filtered through a 0.22 µm filter. On day 1 of differentiation, the culture medium was changed to DMEM/F12 supplemented with 1% P/S, N2-supplement (1:100 dilution), non-essential amino acids (NEAA) (1:100 dilution), NT-3 (10 ng/ml), BDNF (10 ng/ml), doxycycline (4 µg/ml), and mouse laminin (0.2 µg/ml). After 2 days of culturing, we added primary cortical rat astrocytes to the system. One day after astrocyte plating, the culture medium was changed to neurobasal medium supplemented with 1% P/S, B-27 supplement (1:50 dilution), glutamax (1:100 dilution), doxycycline (4 µg/ml), NT-3 (10 ng/ml), and BDNF (10 ng/ml). We stopped proliferation of undifferentiated PSCs and proliferating astrocytes by supplementing the growth medium with 2 µM cytosine arabinoside (Ara-C) on day 3 of cell culture. From day 10 onwards, the neurobasal growth medium was additionally supplemented with 2.5% fetal bovine serum (FBS), to promote the rat astrocytes. iNs were kept for 2–8 weeks in culture with 50% medium changed every 2–3 days (*Figure 1B*).

## Single-cell transcriptomic analysis
### Single-cell RNA-seq data generation

For each cell line, an entire six-well plate was plated with 120.000–150.000 iPSCs and 100.000 astrocytes/well. The cells were differentiated into iNs according to the abovementioned protocol. On d5, d14, d28, and d35, the cells were dissociated into single cells to perform a 10× analysis. All reagents were filtered through a 0.22 µm filter. Each well was washed with DPBS before the cells were incubated for 5 min in 1 ml EDTA 1% accutase/well. Using 1 ml pipette tip, cells were carefully detached from the plate by continuous rinsing to obtain a cell suspension enriched for iNs. All wells of one cell line were pooled and centrifuged for 5 min at 300 g. Then, the supernatant was discarded and the pellet was resuspended in 1 ml accutase with 0.3 µl DNAse (2000 U/µl) for 5 min. The cell pellet was washed two to three times in 5 ml Neurobasal medium supplemented with 10% FCS followed by centrifugation for 5 min at 300 g. Then, the pellet was pipetted 20 times until complete resuspension in 20–00 µl Neurobasal medium supplemented with 10% FCS and filtered through a 30 µm filter. Finally, we centrifuged cells on a Percoll gradient for 5 min at 300 × g to enrich for healthy cells and remove debris. The Percoll gradient consisted of three layers: Lower layer: 4 ml wash medium and 1 ml Percoll solution; middle layer: 4 ml wash medium and 0.75 ml Percoll solution; upper layer: 4 ml wash medium and 0.5 ml Percoll solution. Cells were analyzed using Trypan Blue assay and counted with the Automated Cell Counter Countess (Thermo Fisher) and diluted to yield a concentration of 500–1000 cells/ml to obtain approximately 3000 cells per lane of a 10× microfluidic chip device.

Human and chimpanzee cells were pooled for the 10× experiments and separated computationally afterwards.

Single-cell gene expression libraries were generated using the 10× Chromium Single Cell 3' v2 or v3.1 Kit following the manufacturer's instructions. Quantification and quality control of libraries was performed using a High Sensitivity DNA chip for Agilent's Bioanalyzer and sequenced on a HiSeq2500 in Rapid sequencing mode.

## Data processing

Chromium single-cell RNA-seq (scRNAseq) output was processed using Cell Ranger pipelines. Raw base call (BCL) files were first demultiplexed into FASTQ files with Cellranger mkfastq. Raw reads were aligned to human hg19 and mouse mm10 reference genomes to separate primate cells versus rat astrocytes. Default values defined by Cellranger counts were used to determine primate and rat cells. After removal of rat astrocyte cells and primate-rat multiplets, scRNAseq runs that included both an ape and a human batch were computationally separated. FASTQ files of a published scRNA-seq dataset from cerebral organoids (*Kanton et al., 2019*) containing the same lines as used in this study were processed equally with Cellranger (v2.1.0) count using GRCh38 as a reference. Cellranger v4.0.0 was used for new sets of cell lines H9, SC102A1, JoC, and BmRNA. The resulting BAM files were processed to the VCF format using SAMtools 1.3.1 and BAMtools 1.4. The obtained polymorphisms were used to demultiplex the cells using demuxlet (*Kang et al., 2018*). Cell tags that were positive for more than one line were discarded as multiplets.

We retained genes and single cells for downstream analysis also based on following quality measures: the mean count in at least one batch was larger than 0.1. Means are defined as the sum of counts for one gene divided by the number of cells for one batch. We removed cells with less than 500 or more than 6000 genes detected and cells in which more than 5% of the transcripts accounted for mitochondrial genes. After filtering, 36,034 cells on culturing d 5, d 14, d 28, and d 35 were retained (*Figure 3A*), and 2474 genes and 7626 UMIs (*Figure 3—figure supplement 1*) were detected in each single cell on average. For human and chimpanzee iPSCs, 3329 cells were retained after filtering, and 2140 genes and 7651 UMIs were detected in each single cell on average.

## Identification of neuronal cells and differentially expressed genes

Data normalization, scaling, and variable feature finding were performed using 'SCTranform' function of *Seurat* R package (v3.2.0) (*Stuart et al., 2019*). Multiple batches of single-cell datasets (a batch is defined as one experiment of differentiation and harvesting of iNs) were integrated with 'FindIntegrationAnchors' and 'IntegrateData' functions of *Seurat* R package with the first 15 components were used for finding the anchors (*Stuart et al., 2019*), and 2000 highly variable genes were obtained, which were used to perform principal component analysis (PCA). The first 19 PCs were estimated significantly enriched with variable genes based on the percentage of variance explained by each one with the 'ElbowPlot' function in the *Seurat* R package and were selected as the significant components for *t*-SNE analysis. We performed clustering of single cells with the parameter *resolution* to 0.4 using 'FindClusters' function in *Seurat*. Clusters with similar marker gene expressions were merged manually. Neuronal cells and progenitors were identified according to their expressions of corresponding marker genes: progenitor markers (*SOX2, VIM, ID3, MKI67, TOP2A, CENPF*) (*Yuzwa et al., 2017*, pan-neuron markers [*MAP2, SYT4, SYT1, STMN2, GAP43, DCX, NEFM, VGLUT2*]) (*Basi et al., 1987*; *Fremeau et al., 2004*), sensory neuron genes (*PRPH, ISL1, BRN3A, PHOX2B, TAC1, GAL, SSTR2, PIEZO2, FGF3, SCN9A, NTRK2*) (*D'Autréaux et al., 2011*; *Zou et al., 2012*), and fibroblast cluster markers (*COL1A1, COL1A2, MYL9, ACTA2, TAGLN*) (*Aldeiri et al., 2017*).

Differentially expressed genes (DEGs) between neuronal cells and neural progenitors of each species at each time point were identified by using the 'FindMarkers' function in the *Seurat* R package with the cut-offs of adjusted p value (Wilcoxon test) less than 0.05 and log2 transformed fold change larger than 0.25. We here focused only on DEGs having higher levels in neuronal cells (neuron-expressed genes).

To compare gene expressions before and after NGN2 induction in differentiated (neurons and progenitors) and iPSCs, expressional datasets were integrated with the same procedure as described above. New cell identities of ape and human iPSCs, neurons, and progenitors were set,

using the 'Idents' function in the *Seurat* R package, and then DEGs between cell identities were identified with the 'FindMarkers' function in *Seurat*.

## Gene ontology enrichment analysis

DEGs (neuron-expressed genes) of each species were used to identify over-represented biological processes (BPs) of gene ontologies (GOs) and the genes detected by scRNAseq in ape or human neurons and progenitors were used as the background, respectively. The human GO database on biological process was download with the 'loadGOTerms' function in the *goSTAG* R package including 18,446 genes in 3599 GO categories of biological processes (*Bennett and Bushel, 2017*). The exact binomial test was used to identify categories of biological processes with an enrichment of DEGs (neuron-expressed genes) with a high fraction compared to the fraction of detected category genes in the background list. The binomial test was performed using the 'binom.test' function in the R package. The significantly enriched GO categories were selected based on the p values being less than 0.05. To calculate the Pearson correlation of GO enrichment (p values) of iNs sub-clusters of humans and apes at all time points, we used the 'rcorr' function in the R package 'Hmisc'. To visualize how DEGs (neuron-expressed genes) regulate synaptic functions at each time point, we perform GO enrichment analysis with the online *SynGO* annotations including 179 synaptic processes based on published, expert-curated evidence (*Koopmans et al., 2019*). Significantly enriched synaptic processes were retained with the FDR less than 0.05.

## iNeuronExplorer: an open-source app to explore evolutionary neurobiology in a dish

To make our scRNAseq data accessible to the neuroscience community, we provide a ShinyApp-based web browser for data exploration, called iNeuronExplorer (*Kanton S, 2021*). Thereby, users can search for expression of their gene of interest in iNs. The ShinyApp provides a rich source of expression data and will hopefully be helpful for neuroscientists interested in evolutionary biology and neurological disorders.

## Electrophysiology

iNeurons were grown onto poly-L-ornithin and mouse laminin-coated coverslips (12 mm diameter, Kleinfeld, Germany). Individual coverslips were carefully transferred to a recording chamber under an upright microscope (Olympus BXW-51, Hamburg, Germany) and were perfused (2–3 ml/min) with an artificial cerebrospinal fluid (aCSF; in mM: 100 NaCl, 3.5 KCl, 1 MgCl$_2$, 2 CaCl$_2$, 30 NaHCO$_3$, 1.25 NaH$_2$PO$_4$, 10 glucose) continuously bubbled with carbogen (95% O$_2$ and 5% CO$_2$) to maintain a pH < 7.4. Cells were visualized under differential-interference contrast and selected for smoothness of membrane surface and three-dimensional, neuron-like appearance. As bipolar neurons were rare, we selected multipolar cells that could easily be recognized by their slightly irregular (triangular, square, or star-like) appearance, caused by the branching neurites. Recordings of single cells were done in the whole-cell configuration of the patch-clamp technique with glass electrodes (4.5–5.5 MΩ; oD/iD, borosilicate, Hilgenberg, Germany), pulled to tip resistances of ~5.5 MO (Sutter, P97) and filled with an internal solution containing 130 mM K-gluconate, 10 mM NaCl, 4 mM Mg-ATP, 0.5 mM GTP, 10 mM HEPES, and 0.05 mM EGTA (adjusted to pH = 7.3; filtered at 0.2 µm). The liquid junction potential (15 mV) was corrected online. Intra- and extracellular solutions were adjusted to 270 and 280 mOsm, respectively. This osmolarity closely matched the cell-culture medium and allowed for long-time recordings with stable series resistances (Rs, was typically 15–20 MΩ). Recordings were carried out at room temperature (22–24°C) using an EPC-10 amplifier (HEKA, Lambrecht, Germany). Acquisition and analysis were performed using Patch and Fitmaster software (Ver. 2.9x, HEKA) in conjunction with Origin (Ver 2018-2019b, OriginLab, Northampton, MA). Recordings were digitized at 10 or 20 kHz and done in either the voltage- or current-clamp configuration. With the very first voltage, -ramp and –step protocols, we identified initial values of input resistance, Vrmp and cell capacitance (see below). We excluded data when these values had changed by more than 20%. Rs was typically 15–20 MΩ and 50% Rs-compensation was applied routinely. Data were excluded when series resistances increased by more than 5 MO or exceeded 25 MO. In the current clamp, the bridge compensation was used to compensate any voltage drop across the series

resistances. A very few cells were not recorded at all as the initial voltage ramp indicated the absence of any voltage-gated currents.

## Recordings

The following basic properties were measured: (1) Vrmp or membrane potential, determined by the concentration of ions on both sides of the membrane, the membrane's permeability and activity of ion pumps; (2) Rcell or cell-input resistances, influenced by the cell size and the number of membrane proteins, for example ion channels and transporters; (3) Ccell or cell capacitance, reflects the membranes characteristic to act as a capacitor. As the capacitance influences the impedance – the resistance to AC currents – it influences the propagation of fast-changing signals, for example APs (4) tau = R*C or resulting time constant, a measure of the cell response to stimuli.

The basic properties of the iNs as cell-input resistances (Rcell) and whole-cell capacitances (Ccell) were obtained from the software and verified using the following two protocols. First, we recorded four successive uncompensated voltage steps from −80 to −95/−85 mV/−75–65 mV (20 kHz sampling rate) to obtain Ccell from the initial fast-capacitive decay and Rcell from the plateau current (for details, see Analysis). Second, we used a ramp protocol (from −90 mV [200 ms] rising to +60 mV within 800 ms; back to −90 mV) to verify the presence of voltage-activated currents and to determine Rcell from the current slope and the resting-membrane potential (Vrmp) from the zero-current crossing of the voltage ramp. Thereafter, we recorded EPSCs at −80 mV for 60 s without pharmacological interference (10 kHz sampling rate, filtered at 2.5 kHz). In few recordings, we applied 50 µM kynurenic acid to verify their glutamatergic identity. We then switched to the current-clamp configuration and measured the induction of action potentials by depolarization. We determined the current ΔI that induced app. 5 mV change at rest (app. −80 mV). We then hold the cell at app. 90 mV to remove any inactivation of voltage-gated ion channels and recorded 20 current steps (500 ms long, two hyperpolarizing, 18 depolarizing) with changing current amplitudes as determined previously (ΔI).

## Analysis

To obtain Ccell from the first protocol, the initial 500 µs of the fast-capacitive decay were fitted with an exponential function. The obtained time-constant tau allowed determination of Ccell (C = Rs/tau). In addition, Ccell was determined using the RC compensation routine from the Patchmaster software, once automatically and once manually under visual control. The plateau currents after the transient decay (10 ms mean after 20 ms) allowed determination of Rcell (Rcell = ΔU/ΔI). A second value we obtained from the ramp protocol using the current slope between 75 and 65 mV (Rcell = ΔU/ΔI), and a third value from the amplifier of the Patchmaster software. Normally, all three values for Ccell and for Rcell were in good agreement (<10%) and were averaged. In case of disagreements, the traces were reanalyzed and the accurate value determined by hand. The resting-membrane potential (Vrmp) was obtained from the zero-current crossing of the voltage ramp. In some cases where a low subthreshold activation of voltage-gated sodium currents hyperpolarized the membrane close to and before the zero-current crossing, the resting potential was estimated using a linear fit to the slope between −75 and −65 mV and determining its zero crossing. Action potentials in the current clamp protocols were determined as positive crossings at −20 mV. The maximal numbers obtained during these steps were used for analysis. EPSCs traces were exported for further analysis to the program NeuroExpress (NeXT, v. 19 .n; Attila Szücs, UCSD, San Diego, CA) via an Igor binary format. There we used the 'Triangular template' with the following settings for an analysis of single excitatory events: estimated noise: 14 pA, max EPSC rise-time: 1.5 ms, max EPSC fall time: 6 ms, max EPSC amplitude: 500 pA.

## Lipofection of iNeurons

To generate GFP-expressing iNs, cells were lipofected 4 days before fixation using the Lipofectamine 3000 Transfection Kit from Invitrogen. Briefly, 5 µl OptiMEM was mixed with 0.75 µl lipofectamine and then added to 25 µl OptiMEM containing 0.1 µg GFP pMAX plasmid (Lonza) and 1 µl P3000 reagent. After incubation for 15 min, 50 µl lipofection mix was added per coverslip, and the iNs were kept in culture until fixation.

**Immunostaining of iNeurons**

## Preparation of paraformaldehyde fixative

To prepare 100 ml 4% paraformaldehyde, all steps were performed under the hood. Four gram PFA was added to 40 ml distilled water and was heated to exactly 60°C while stirring. Then, 1 N NaOH was added dropwise until the solution cleared. Four gram sucrose and 50 ml 240 mM Na phosphate buffer (pH 7.4) were added to the mixture. After cooling, the pH of the solution was checked and, if necessary, adapted to pH 7.4 by adding HCl. The solution was brought to a final volume of 100 ml by adding bidistilled $H_2O$ and then filtered through a 0.22 µm filter. Five to ten milliliters aliquots was stored at $-20$°C.

## Fixation of GFP-labeled iNeurons

GFP-pmax-lipofected iNs were fixed at days 7, 14, 21, and 35 of differentiation. The coverslips were transferred with tweezers from the 24-well cell culture plate into a tissue culture dish (35 mm) containing 1 ml PBS. One milliliter 4% PFA (pre-warmed to 37°C) was added to the dish under a hood, and the cells were incubated for 8 min. The coverslips were washed with PBS three times and stored in PBS at 4°C. The coverslips were blinded after the fixation step, before the start of the immunostaining.

## Quenching and immunostaining of GFP-labeled iNs
### Analysis of GFP-labeled iNs direct fluorescence

PBS was aspirated, and the coverslips were incubated in 1 ml 0.2 M glycine buffer for 30 min. After three quick washes with immunofluorescence buffer (IF-buffer: 120 mM phosphate buffer cointaining 0.2% gelatin and 0.05% Triton X-100), the coverslips were mounted on slides with Mowiol 4–88 and DAPI (1:1000 dilution) and stored at 4°C.

### Immunostaining of GFP-labeled iNeurons

iNs were permeabilized with 0.05% Triton X-100 in PBS for 10 min and quenched in 1 ml 0.2 N glycine buffer for 30 min at room temperature. The coverslips were washed with PBS and incubated with 100 µl of the respective primary antibody diluted in IF-buffer (see *Supplementary file 2* for the list of Abs). The coverslips were washed with IF-buffer five times for 5 min and then incubated with 100 µl of the respective secondary antibody diluted in IF-buffer containing DAPI (dilution 1:1000). After five washes with IF-buffer for 5 min and three quick washes with PBS, the coverslips were mounted on slides with Mowiol 4–88 and stored at 4°C.

Image Acquisition iNs were acquired as confocal Z-stacks. iNs were imaged using an Olympus FV1200 confocal microscope equipped with a 40× oil immersion objective (optical section thickness: 1.028 µm, distance between consecutive optical sections: 0.4 µm) or a Zeiss LSM 780 NLO two-photon upright microscope equipped with a 63× Neofluar immersion objective (optical section thickness: 0.8 µm, distance between consecutive optical sections: 0.6 µm). The acquisition of day 35 data set was carried on using a spinning disk Andor IX 83, inverted stand, equipped with a 60× oil immersion objective. We acquired three-dimensional (Z-stack) tile scans with a number of z-sections ranging from 5 to 30 depending on the cell. Single tiles were 1024 × 1024 pixels. The following lasers were used: 405 nm for DAPI, 488 nm for GFP, and 561 nm for RFP/Alexa-555. The acquisition experiments were performed in blind.

**Image quantification**

## Quantification of neuronal morphology

We quantified GFP-expressing iNs at d7, 14, 21, and 35 of differentiation (see *Supplementary file 1* for a summary of the number of cells traced). Three-dimensional confocal Z-stacks were traced and analyzed using Imaris9.5 (*Bitplane, 2020*). We used a manual tracing approach with a neurite's diameter for of 0.4 µm. For quantification, we exported Sholl intersections and .hoc files, as they contain information about both the neurite length and the branching pattern. HOC files were analyzed using a custom script (*Peter and Schörnig, 2020*). Statistical analyses were performed using R version 3.4.4. Experiments were unblinded only after running the complete quantifications. Note that the increase in neurite length is batch dependent for both bipolar and multipolar cells (a batch is defined as one experiment of differentiation of PSCs into iNs; bipolar cells: $F_{(1,220)} = 13.1790$,

$p_{batch} < 0.0012$ TwoWayAnova; multipolar $F(1,454) = 17.2302$, $p_{batch} < 0.001$). In *Figure 2*, we normalized for the batch effect by dividing the data for length of the longest neurite by the mean of the ape data per batch.

## Quantification of TuJI signal

In *Figure 1—figure supplement 1*, the images of the time course of TUJI expression were acquired sequentially using the same settings during one session at the Zeiss LSM 780 NLO microscope. For each image, we use FiJI and quantify the signal in an area containing only neurites (but not cell bodies). The signal is expressed as arbitrary units (AU).

## Assignment of cell identity

In this study, we define sensory and cortical neurons based on marker expression at the mRNA (scRNAseq) and protein level (immunofluorescence). Thereby, we define sensory neurons as neurons expressing sensory neuron markers. We used intermediate neurofilament peripherin (*PRPH*) (*Liem and Filaments, 2013*), neurotrophin receptor tyrosine receptor kinase B (*TRKB*, encoded by *NTRK2*), homeodomain transcription factors (*POU4F1/BRN3A*), paired-like homeobox 2b (*PHOX2B*) (*D'Autréaux et al., 2011*; *Zou et al., 2012*), and the LIM homeodomain transcription factor Islet1 (*ISL1*) (*Sun et al., 2008*) as sensory neuron markers.

N number of ape and human unipolar, bipolar, and multipolar iNs at d7, 14, 21, and 35 of differentiation. iNs were traced using image analysis software (Imaris) and quantified using a custom software (*Peter and Schörnig, 2020*) for morphological analysis.

Electrical properties were measured for the resting-membrane potential (Vrmp), cell resistance (Rcell), the cell capacitance (Ccell), the time constant (Tau), the number of actional potentials (APs), and spontaneous excitatory synaptic currents (EPSCs). Shown is for the different species on d2 to >d49 of differentiation the n number of cells analyzed, the median, mean, SD and SMEM values and p values for an unpaired t-test with welch correction for pairwise comparison of APs and EPSCs between the species and p values for an anova analyses on age and species effects and interaction of species and age. Bold values represent significant p values.

Statistics are shown for the neurite length of bipolar and multipolar iNs, the relative longest neurite (axon) length of multipolar and bipolar iNs, the relative number of sholl intersections for multipolar and bipolar iNs, and the total dendrite length of multipolar and bipolar iNs. Shown is for the species on d7, d14, d21 and d35 of differentiation the n number of cells analyzed, the differences between species in %, the mean, SD and the p values for pairwise comparisons (Mann–Whitney U test/unpaired t-test). The differences in % between the species are indicated with '–' for a difference where ape iNeurons show higher values and '+' for differences where the human iNs show higher values. Additionally, we show the values for a Shapiro test, and a two-way anova on the interaction of species and day. Significant p values are shown in bold.

Statistics are shown for the number of neurons expressing BRN2, ISL1, CUX1, PRPH, TBR1, and NAV1.7 protein. A Tukey's post hoc test revealed no significant pairwise differences between the species.

## Acknowledgements

The authors thank Svante Pääbo for the continuous support and discussion on the project. We thank Takashi Namba, Nereo Kalebic and Samir Vaid for discussion and input. We thank Kathrin Köhler, Linda Dombrowski, Katrin Linda, Teun Klein Gunnewiek, Noor Smal and Malgorzata Santel for excellent technical help, support and advice, the Tchimpounga Sanctuary for support with usage of the JoC iPSC line and the Max-Delbrück-Centrum für Molekulare Medizin, Berlin for providing the chimpanzee iPSC line Chimp male iPSC Sendai CL5 (ciPS01), the light microscopy facility at the MPI-CBG in Dresden, especially Sebastian Bundschuh and Britta Schroth-Diez, for help and assistance with confocal microscopy. Experiments and generation of NGN2-inducible PSC lines were in part performed at Radboudumc, Nijmegen, The Netherlands. This work was supported by the Max Planck Society.

# Additional information

## Funding

No external funding was received for this work.

## Author contributions

Maria Schörnig, Conceptualization, Software, Formal analysis, Writing - original draft, Project administration, Writing - review and editing, MS generated Ngn2/rtTA positive pluripotent stem cell lines and grew iNs cultures, performed immunohistochemical staining and microscopy. MS wrote the custom script for morphological analyses. MS performed morphological analyses. MS performed single cell RNAseq experiments. MS designed the study. MS wrote the manuscript with support from all authors; Xiangchun Ju, Formal analysis, Writing - original draft, Writing - review and editing, XJ analyzed single cell RNAseq data. XJ wrote the manuscript with support from all authors; Luise Fast, Investigation, LF assisted with iNs cultures. LF performed immunohistochemical staining and microscopy. LF performed morphological analyses. LF performed single cell RNAseq experiments; Sebastian Ebert, Formal analysis, SE performed single cell RNAseq experiments. SE analyzed single cell RNAseq data; Anne Weigert, Formal analysis, Investigation, AW assisted with iNs cultures. AW generated and characterized the human iPS cell lines HmRNA and HmRNA_Ngn2 and the chimpanzee iPS cell line ciPS01_Ngn2. AW performed morphological analyses. AW performed single cell RNAseq experiments; Sabina Kanton, Data curation, Formal analysis, Visualization, Writing - review and editing, SK designed the ShinyApp-based web browser; Theresa Schaffer, Investigation, TS assisted with iNs cultures; Nael Nadif Kasri, Resources, Writing - review and editing, NNK provided information and material relevant for the iNs generation and interpretation of the data; Barbara Treutlein, Conceptualization, Supervision, BT designed the study; Benjamin Marco Peter, Software, BP wrote the custom script for morphological analyses; Wulf Hevers, Investigation, Visualization, Methodology, Writing - original draft, Writing - review and editing, WH performed and analyzed electrophysiological experiments. WH wrote the manuscript with support from all authors; Elena Taverna, Conceptualization, Data curation, Supervision, Visualization, Methodology, Writing - original draft, Project administration, Writing - review and editing, ET performed immunohistochemical staining and microscopy. ET designed the study. ET wrote the manuscript with support from all authors

## Author ORCIDs

Maria Schörnig ⓘ https://orcid.org/0000-0001-5334-5342
Benjamin Marco Peter ⓘ https://orcid.org/0000-0003-2526-8081
Wulf Hevers ⓘ http://orcid.org/0000-0003-1881-5913
Elena Taverna ⓘ https://orcid.org/0000-0002-2430-4725

## Ethics

Human subjects: For this study we used three human (hiPS409-B2, SC102A1, HmRNA), three chimpanzee (SandraA, JoC, ciPS01) and one bonobo (BmRNA) iPS cell lines and one additional human ES cell line (H9). The human iPSC lines hiPS409-B2 and SC102A1 were purchased from the Riken BRC Cellbank and System Biosciences, respectively. The human iPSCs line HmRNA (generated in this study) was reprogrammed from human dermal fibroblasts using the StemMACS mRNA transfection kit. The cell line was validated for pluripotency markers by immunohistochemical staining using the Human Pluripotent Stem Cell 3-Colour Immunohistochemistry Kit and were differentiated into the three different germ layers using the Human Pluripotent Stem Cell Functional Identification kit and StemMACS Trilineage Differentiation Kit. Karyotyping was carried out using Giemsa banding at the Stem Cell Engineering facility, a core facility of CMCB at Technische Universität Dresden. Karyotypes were found to be normal. The human ES cell line H9 was purchased from WiCell. The chimpanzee iPSC lines SandraA and JoC as well as the bonobo iPSCs line BmRNA were generated in a previous study (Kanton et al., Nature, 2019). The chimpanzee iPSCs ciPS01 line was provided by the Max-Delbrück-Centrum für Molekulare Medizin, Berlin. The rtT A/Ngn2-positive iPSCs/ESCs hiPS409-B2_Ngn2, SandraA_Ngn2, BmRNA_Ngn2, H9_Ngn2, SC102A1_Ngn2, HmRNA_Ngn2, ciPS01_Ngn2 and JoC_Ngn2 were generated using lentiviral vectors to stably integrate the transgenes into the genome of the stem cells and differentiate the stem cells into neurons as previously described by

Frega et al., Jove, 2017. Our cultures were regularly controlled for mycoplasma. Permission to work with human and non-human primate iPSC lines and Ngn2-inducible cell lines was obtained through the Sächsisches Staatsministerium für Umwelt und Landwirtschaft (Az.: 55-8811.72/26, Az.: 55-8811.72/26/350). The use of human ESCs was approved by the ethics committee of the Robert Koch Institut (https://www.rki.de/DE/Content/Gesund/Stammzellen/Register/reg-20161027-Paeaebo.html).

## Decision letter and Author response
Decision letter https://doi.org/10.7554/eLife.59323.sa1
Author response https://doi.org/10.7554/eLife.59323.sa2

# Additional files
## Supplementary files
- Supplementary file 1. N number of iNs used for morphological analysis.
- Supplementary file 2. Materials.
- Supplementary file 3. Statistics on the electrical properties.
- Supplementary file 4. Statistics on the morphological analysis.
- Supplementary file 5. Overview on iNs batches and number of cells analyzed for scRNAseq, electrophysiological and morphological experiments.
- Supplementary file 6. Statistics on the neuronal subtypes.
- Transparent reporting form

## Data availability
Sequencing data for single cells have been deposited in ArrayExpress under the accession code E-MTAB-9233 and under Mendeley Data with doi: 10.17632/y3s4hnyvg6. To make our scRNAseq data accessible to the neuroscience community, we provide a ShinyApp-based web browser for data exploration, called iNeuronExplorer (https://bioinf.eva.mpg.de/shiny/iNeuronExplorer/). Morphological data for neurons and a custom made script for analysis have been deposited in GitHub under the URL: https://github.com/BenjaminPeter/schornig_ineuron; copy archived at https://archive.softwareheritage.org/swh:1:rev:99e78f21b625d637acc871bf43bd75f5af621288.

The following datasets were generated:

| Author(s) | Year | Dataset title | Dataset URL | Database and Identifier |
|---|---|---|---|---|
| Ju X, Schörnig M, Ebert S, Treutlein B, Taverna E | 2020 | scRNAseq dataset | https://www.ebi.ac.uk/arrayexpress/experiments/E-MTAB-9233/ | ArrayExpress, E-MTAB-9233 |
| Peter B, Schörnig M | 2020 | Scripts for Schoernig et al. 2020 | https://github.com/BenjaminPeter/schornig_ineuron | BenjaminPeter / schornig_ineuron, BenjaminPeter / schornig_ineuron |
| Kanton S | 2020 | iNeuronExplorer | https://bioinf.eva.mpg.de/shiny/iNeuron_Explorer/ | MPI EVA webbrowser, shiny/iNeuron_Explorer/ |

The following previously published dataset was used:

| Author(s) | Year | Dataset title | Dataset URL | Database and Identifier |
|---|---|---|---|---|
| Lin HC, He Z, Ebert S, Schörnig M, Santel M, Weigert A, Hevers W, Nadif Kasri N, Taverna E, Camp JG, Treutlein B | 2020 | scRNAseq dataset | https://dx.doi.org/10.17632/y3s4hnyvg6 | Mendeley Data, 10.17632/y3s4hnyvg6 |

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
