## [Decision Letter]

**Acceptance summary:**

Your work presents an elegant comparison of structural and functional maturation of cortical neurons from different primate species that is of broad interest to researchers interested in evolutionary neuroscience and those who are interested in the unique qualities of the human cortex.

**Decision letter after peer review:**

Thank you for submitting your article "Comparison of induced neurons reveals slower structural and functional maturation in humans than in apes" for consideration by *eLife*. Your article has been reviewed by three peer reviewers, including Anita Bhattacharyya as the Reviewing Editor and Reviewer #1, and the evaluation has been overseen by Piali Sengupta as the Senior Editor.

The reviewers have discussed the reviews with one another and the Reviewing Editor has drafted this decision to help you prepare a revised submission.

The manuscript by Schörnig presents an elegant comparison of structural and functional maturation of cortical neurons from different primate species that is of broad interest to researchers interested in evolutionary neuroscience and those who are interested in the unique qualities of the human cortex. The authors use an induced neuron approach to generated cortical like neurons from iPSCs form different species and compare the structure, function and gene expression of the different neuron over time in culture. This strategy bypasses development and provides much more heterogeneous cultures for analysis. While the results are largely descriptive, they provide very interesting resource data providing insight into both primate neural development and human-specific attributes.

While the reviewers agree that the manuscript has potential, extensive revisions and new data are needed. Specifically, three points that require additional support:

1) Definitive characterization of sensory neurons

The identity of the induced neurons as sensory neurons is interesting but is based solely on gene expression and clustering of the scRNAseq data. To conclude that these neurons are indeed sensory neurons, the authors need cellular characterization (ephys, marker expression). To tease apart whether the cultures are different than previous studies, the authors need to classify the scRNA data from previous work utilizing NGN2 induction (Zhang et al., 2013) with this same protocol.

2) Rigor of experimental design

The comparisons of species differences by scRNA-seq lack rigor based on the use of only a single individual from one species. In addition, there are no details of how many batches of differentiation/induction were done or how many replicates were used for analysis. These points need to be addressed in revision.

Heterogeneity of cultures and composition differences between species needs to be taken into account to support claims that the molecular, morphological, and physiological timing differences are explained by species and not cell type. It is crucial to compare cells of the same type for each analysis to rule out the influence of composition differences for any of the species differences claims.

3) Conclusions and mechanisms

Results in sensory neurons links to working memory in the prefrontal cortex is not well supported.

The manuscript does not build on the rich and extensive transcriptomic data to provide any mechanistic hypotheses of the causes of the differences.

[Editors' note: further revisions were suggested prior to acceptance, as described below.]

Thank you for resubmitting your work entitled "Comparison of induced neurons reveals slower structural and functional maturation in humans than in apes" for further consideration by *eLife*. Your revised article has been evaluated by Piali Sengupta (Senior Editor) and a Reviewing Editor.

The revised manuscript has been strengthened by additional data including better characterization of the resulting sensory neuron population and the composition similarities between cultures of ape and human. Additional analyses of the transcriptomic data strengthens the overall impact of the data.

There are some remaining issues that need to be addressed before acceptance, as outlined below:

1) The organization of the manuscript would flow better if the data were presented in this order: morphological, transcriptomic and functional data. Thus, the data in Figure 5 should be moved after Figure 1. The data in Figure 5—figure supplement 2 would be useful to include in the main figure.

2) Statistics for structural/morphological analyses are not provided. Please add to text or Figure legend.

Figure 2D

Figure 2—figure supplement 4

Figure 2—figure supplement 5

3) While the characterization of the iNs as primarily sensory neurons is better supported, the authors have not shown that the functional and morphological phenotypes are attributable solely to sensory neurons. The authors should consider removing sensory neurons from the Abstract, first sentence.

---

## [Author Response]

The manuscript by Schörnig presents an elegant comparison of structural and functional maturation of cortical neurons from different primate species that is of broad interest to researchers interested in evolutionary neuroscience and those who are interested in the unique qualities of the human cortex. The authors use an induced neuron approach to generated cortical like neurons from iPSCs form different species and compare the structure, function and gene expression of the different neuron over time in culture. This strategy bypasses development and provides much more heterogeneous cultures for analysis. While the results are largely descriptive, they provide very interesting resource data providing insight into both primate neural development and human-specific attributes.

We thank the reviewers for appreciating our work and effort.

Based on their comments and constructive criticisms, we designed additional experiments and further analysis that -we think- expanded and strengthened our findings. Following the philosophy of data sharing, we developed and added to the present revised version a Shyny-App called iNeuronExplorer to share our datasets with the entire neuroscience and stem cell community. Note that we updated the authors list to include Sabina Kanton, who developed iNeuronExplorer (see also authors contribution).

While the reviewers agree that the manuscript has potential, extensive revisions and new data are needed. Specifically, three points that require additional support:

To address the reviewer’s comments and concerns, we performed further characterization using (i) fluorescence immunostaining for fate markers (ii) electrophysiology and (iii) single cell RNA sequencing on additional lines to assess the differences in composition of the different stem cell lines and neuronal populations.

Below, we describe in detail new experiments, new analysis and new tools development implemented during the revision.

1) Definitive characterization of sensory neuronsThe identity of the induced neurons as sensory neurons is interesting but is based solely on gene expression and clustering of the scRNAseq data. To conclude that these neurons are indeed sensory neurons, the authors need cellular characterization (ephys, marker expression).

We totally agree on the concept and need of using different means of assessing cell fate, other than scRNAseq.

We therefore performed immunostaining for peripheral, cortical and sensory markers on 2 human, 1 chimpanzee and 1 bonobo samples on day 21 and day 35 of differentiation. The data are included in two new Figure Supplements, Figure 3—figure supplement 4 and Figure 3—figure supplement 5.

The new Figure 3—figure supplement 4 shows data regarding:

- the bona fide peripheral marker PRPH

- the deep layers marker TBR1

- the upper layers markers CUX1 and BRN2

The new Figure 3—figure supplement 5 shows data regarding:

- the sensory marker ISLET1

- the sensory marker NAv1.7

To provide a comprehensive picture of iNs fate, for all markers considered we show:

- the tSNE plot of the given marker

- the quantification of expression in iNs for that given marker, based on scRNAseq data

- a representative immunofluorescence image of the given marker

- the quantification of the % of iNs immune-positive for that given marker

The results from the new experiments revealed that

i) the iNs culture is heterogeneous,

ii) the vast majority of the iNs are upper layers sensory cortical neurons

iii) a minor proportion of iNs are peripheral

iv) despite all this heterogeneity, each species had similar proportions of peripheral, cortical and sensory neurons, indicating that there was no particular preference for a given cell type between the species.

Of note, the presence of iNs double positive for ISLET1 and BRN2 (mRNAand protein level) show in a more direct way the presence of cortical sensory fate.

Based on these data, we conclude that at least a proportion of iNs might be high-order sensory neurons of the cortex.

A surprising finding in our work is the presence of a vast proportion of sensory neurons, something that was under-appreciated in previous reports.

To gain functional insight in the sensory fate, we tested a subtype-differentiating Na-channel blocker PF‑05089771 as probe for sensory neurons. The three main Na-channels indicated by our scRNAseq data would show either a very high- (Nav1.7/SCN9a ~ 0.011uM), a medium- (Nav1.2/SCN2a ~ 0.11uM) or low sensitivity (Nav1.3/SCN3a ~ 11uM) towards this substance (half-inactivated channels, when expressed in HEK293 cells (Brown et al., 2016). As we could not identify a major contribution of Nav1.7 to the voltage-gated sodium currents we observed (only concentrations above 1uM clearly inhibited those currents in iNs (e.g. 10uM ~ 50% inhibition)), we have not further followed this line.

Of note, we observed a clear discrepancy in the % of marker-expressing cells at the protein and mRNA levels. This is not surprising, as it is well known that protein and mRNA levels are not necessarily linearly correlated. Our data would therefore suggest that for certain genes (like *CUX1*) mRNA and protein expression are differentially regulated. In addition, we are aware that technical reasons can also contribute to these discrepancies, like different tendency of different population to be recovered or survived the scRNAsq procedure.

In line with the point originally raised by the reviewer, an important conclusion/caveat that emerges from our work, is the need to always combine mRNA expression, immunofluorescence and functional analysis to assess cell fate in a more comprehensive and coherent way.

To tease apart whether the cultures are different than previous studies, the authors need to classify the scRNA data from previous work utilizing NGN2 induction (Zhang et al., 2013) with this same protocol.

To answer this question, we assessed the mRNA expression of all marker genes found in the single-cell quantitative RT-PCR analysis (Fluidigm) of Zhang et al.

We could identify expression of almost all markers in all species and cell lines, with absence of expression of few genes in our dataset including forebrain marker *FOXG1* and GABA receptor *GABRB2*.

Besides these minor differences that might arise from technical differences between the two studies, we observe the same cortical and excitatory neuron markers as described by Zhang et al.

Expanding the finding of Zhang et al., we find that iNs express also sensory markers, suggesting a sensory identity that was not reported before. We think the sensory identity might have been underrepresented in bulk sequencing data by Zhang et al.

Of note, the sensory fate we report here was confirmed in while we were in revision by two independent laboratories using human iPSCs-derived neurons (Maimaitili et al., 2020; He et al., 2020).

2) Rigor of experimental designThe comparisons of species differences by scRNA-seq lack rigor based on the use of only a single individual from one species. In addition, there are no details of how many batches of differentiation/induction were done or how many replicates were used for analysis. These points need to be addressed in revision.Heterogeneity of cultures and composition differences between species needs to be taken into account to support claims that the molecular, morphological, and physiological timing differences are explained by species and not cell type. It is crucial to compare cells of the same type for each analysis to rule out the influence of composition differences for any of the species differences claims.

To address the questions, we generated scRNAseq data from two additional human (SC102A-1 and H9), one chimpanzee (Jo_C) and one bonobo (BmRNA) iPSC cell lines on day 14 (d14) and day 35 (d35) of differentiation. Thereby we increased the dataset from 25,707 to 36,034 analyzed cells. We chose the cell lines for scRNAseq to match to the cell lines used for morphological and electrophysiological experiments. The timepoints for additional scRNAseq were based on the time point with the biggest differences in maturation of iNeurons between the species (d14) and a timepoint with minor differences and mature neurons (d35). All figures were update accordingly.

After data processing, we retained in total 10,111 ape iNs from the SandraA, JoC and BmRNA cell lines and 25,923 human iNs from the hiPS-409-B2, SC102A-1 and H9 cell lines (see Figure 3A-C). Including the data from four additional cell lines, we still identified 8 clusters of cells based on gene expression patterns (see Figure 3B). Ape and human cells from all cell lines occurred in all three neuronal sub-clusters, with minor differences in proportion (see Figure 3D). Based on these data, we conclude that NGN2 induction in stem cells leads to the differentiation into the same neuronal sub-types, independent of the cell line and species of choice.

To compare cells of the same sub-class for each analysis, we identified genes that are significantly higher expressed in the three iNs sub-populations than in the cells expressing progenitor markers (NPs) at the same time point in the same species and looked for statistical enrichment of these genes among groups of genes assigned to “Biological Processes” in Gene Ontology (Bennett and Bushel, 2017) (Figure 4A;C).

We updated the Figure 4 accordingly:

Figure 4A: Higher expressed genes of iNs sub-clusters at the same time point in the same species.

Figure 4B: Correlation of enriched GO terms in iNs sub types.

Figure 4C: Categories of GO terms on biological processes enriched with higher expressed genes in ape and human iNs sub-clusters at each time point.

Figure 4D: Schematic representation of synaptic parts depicted by SynGO ontology analysis.

Figure 4E: Sunburst plots of enriched synaptic GO terms with higher expressed genes in developing ape and human iNs at each time point.

Figure 4 Figure Supplement 1: Sunburst plots of enriched synaptic GO terms with higher expressed genes conserved in developing ape and human iNs sub-clusters at each time point.

The number of batches and cell lines used is listed in the transparent form, material methods sections and supplementary information. To further address this comment, we generated an additional overview (Supplementary file 5) with all batches (“batch” is the start of new cell differentiation) and the respective number of cells analyzed from each cell culture experiment.

We hope we have addressed the crucial point raised by the reviewer in a satisfactory manner.

3) Conclusions and mechanismsResults in sensory neurons links to working memory in the prefrontal cortex is not well supported.

The general link between high-order sensory neurons and working memory is based on papers present in the literature (see the work of Damasio, Damasio and LeDoux). In the revised form (i) we added more citations to make this point clear and (ii) we re-phrases the sentence in the Discussion, to make clear we are referring to data from others.

The possible link between the iNs we observed in the dish to the high-order sensory neurons in the prefrontal cortex was proposed in the discussion as a speculation from our side. In the present revised version, to corroborate this suggestion, we run double immunofluorescence for sensory (ISL1) and cortical (CUX1, BRN2) markers and show the presence of a population of sensory cortical iNs. These data suggest that at least some iNs must be higher order sensory neurons.

We could not show directly the prefrontal identity as: (i) we do not have markers identifying specifically only prefrontal cortex neurons, and (ii) to the best of our knowledge, scRNAseq data from sensory neurons of the human prefrontal cortex are at the moment underrepresented.

Give these experimental limitations, we did re-phrase the sentence to make the assignment of the prefrontal identity as a possibility rather than a fact.

We thank the reviewer for raising this concern, and we hope we have addressed that in a satisfactory manner.

The manuscript does not build on the rich and extensive transcriptomic data to provide any mechanistic hypotheses of the causes of the differences.

We appreciate the suggestion of the reviewer and decided (i) to proceed to a deeper analysis of the scRNAseq data (ii) to provide speculations and mechanistic hypothesis in the Discussion.

We hope our speculations will meet the reviewers’ expectations.

[Editors' note: further revisions were suggested prior to acceptance, as described below.]

There are some remaining issues that need to be addressed before acceptance, as outlined below:1) The organization of the manuscript would flow better if the data were presented in this order: morphological, transcriptomic and functional data. Thus, the data in Figure 5 should be moved after Figure 1. The data in Figure 5—figure supplement 2 would be useful to include in the main figure.

We re-organized the manuscript according to the suggestions, in the order: morphological, transcriptomic and functional data. The Figures and Figure supplements are restructured and named accordingly. We included the data for total neurite length, axon length, sholl intersections and dendrite length into the new main Figure 2.

2) Statistics for structural/morphological analyses are not provided. Please add to text or Figure legend.Figure 2DFigure 2—figure supplement 4Figure 2—figure supplement 5

We provide the n-numbers in the figure legends and in an additional Supplementary File 6. As a pairwise comparison we performed a Tukey post-hoc test, that revealed no significant pairwise differences of the neuronal sub-types between the species.

3) While the characterization of the iNs as primarily sensory neurons is better supported, the authors have not shown that the functional and morphological phenotypes are attributable solely to sensory neurons. The authors should consider removing sensory neurons from the Abstract, first sentence.

We agree and rephrased the sentence accordingly.